# Diversity and potential host-interactions of viruses inhabiting deep-sea seamount sediments

Meishun Yu[1,3], Menghui Zhang[1,3], Runying Zeng[1], Ruolin Cheng[1], Rui Zhang[2], Yanping Hou[1], Fangfang Kuang[1], Xuejin Feng[1], Xiyang Dong [1], Yinfang Li[1], Zongze Shao[1] ✉ & Min Jin [1] ✉

Seamounts are globally distributed across the oceans and form one of the major oceanic biomes. Here, we utilized combined analyses of bulk metagenome and virome to study viral communities in seamount sediments in the western Pacific Ocean. Phylogenetic analyses and the protein-sharing network demonstrate extensive diversity and previously unknown viral clades. Inference of virus-host linkages uncovers extensive interactions between viruses and dominant prokaryote lineages, and suggests that viruses play significant roles in carbon, sulfur, and nitrogen cycling by compensating or augmenting host metabolisms. Moreover, temperate viruses are predicted to be prevalent in seamount sediments, which tend to carry auxiliary metabolic genes for host survivability. Intriguingly, the geographical features of seamounts likely compromise the connectivity of viral communities and thus contribute to the high divergence of viral genetic spaces and populations across seamounts. Altogether, these findings provides knowledge essential for understanding the biogeography and ecological roles of viruses in globally widespread seamounts.

Seamounts can be both isolated and clustered and are ubiquitous and prominent features of the world's underwater topography, thus forming one of the major biomes of the ocean[1]. The geographic features of seamounts exert complex effects on oceanic circulation and mixing at a scale ranging from regional to more local effects[2]. Interactions between seamounts and steady and variable flows have been described, providing a better perspective for understanding the mechanisms underlying processes that influence biology[3]. As unique ecosystems in the deep ocean, seamounts are generally considered oases of biomass abundance and hotspots of species richness[1,4]. Previous studies on seamount fauna showed that seamounts have a diverse trophic architecture and tend to support aggregations of higher consumers, such as fish[4].

So far, most of our knowledge on seamount biodiversity is derived from studies on seamount fauna, while the diversity and ecology of microbial communities are much less understood in general. Over the past decades, with the application of metagenomics, significant efforts have been made to explore the diversity, function, and ecology of the prokaryotes inhabiting seamount environments[5–9]. For example, Jacobson Meyers et al.[7] explored the extracellular enzyme activity and microbial diversity on seafloor exposed basalts from Lōʻihi Seamount; they suggested that prokaryotes on basaltic rock play a substantial and quantifiable role in benthic biogeochemical processes through transforming organic matter[6]. Huo et al. utilized fosmid sequencing to explore the ecological functions of microbes in a sediment sample collected from the cobalt-rich ferromanganese crust of a seamount

[1]State Key Laboratory Breeding Base of Marine Genetic Resource and Southern Marine Science and Engineering Guangdong Laboratory (Zhuhai), Third Institute of Oceanography, Ministry of Natural Resources, Xiamen 361000, China. [2]Institute for Advanced Study, Shenzhen University, Shenzhen, Guangdong, China. [3]These authors contributed equally: Meishun Yu, Menghui Zhang. ✉e-mail: shaozongze@tio.org.cn; jinmin@tio.org.cn

region in the central Pacific[9]. They suggested that microbes are involved in the nitrogen cycle, and a high frequency of horizontal gene transfer events, as well as genomic divergence, contributed to the adaption of microbes to their deep-sea environment. Collectively, these studies have suggested that the prokaryotes in seamount ecosystems have extensive diversity and play important roles. However, little is known about viral communities and their roles in seamount ecosystems.

Viruses are the most abundant and ubiquitous biological entities on the planet. The vast majority of environmental viruses are phages that infect bacteria[10,11]. Because of their enormous abundance and genetic diversity, viruses are major players in marine ecosystems: (1) Viruses control host abundance and affect the host community structure by killing hosts[12]. (2) Viruses influence host diversity, evolution, and environmental adaptation through horizontal gene transfer, resistance selection, and host metabolism programming[13–15]. (3) Viruses drive biogeochemical cycling by releasing intracellular organic matter from hosts and promoting the transformation of particulate organic matter to dissolved organic matter[16,17]. (4) Viruses assist microbial-mediated biogeochemical cycling processes by expressing auxiliary metabolic genes (AMGs)[18,19].

Currently, there are significant gaps in our knowledge regarding the community structure, genetic diversity, and ecological roles of viruses in seamount ecosystems. To date, only one publication used epifluorescence microscopy to count virus-like particles (VLPs) in deep-sea sediments around two seamounts in the Tyrrhenian Sea. The results showed that benthic viral production was much higher in sediments around seamounts than in non-seamount sediments[20]. Moreover, only one culturable virus has been isolated from seamount environments to date[21]. In view of the central roles viruses play in shaping host communities and mediating biogeochemical cycles, exploring viral communities in seamount ecosystems is essential. In addition, the question of whether seamounts are isolated habitats with highly endemic faunas and related questions of connectivity have been the subject of many studies over the last 30 years[1,3,4]. Overall, scientists have concluded that seamounts do not generally support high levels of endemism[3]. However, this conclusion has been challenged by studies on certain fauna taxa whose life history is characterized by poor dispersal and thus showing low connectivity between seamounts with high endemism at a local level[22]. In this context, research on the effects of geographic features of seamounts on the resulting viral community may also offer novel insights into this controversial topic as well as deepen our understanding of underlying mechanisms regarding the generation and maintenance of local viral diversity.

Here, we utilized high-depth sequencing and viral-sequence specific bioinformatics tools to explore the diversities, biogeography, and potential ecological roles of viruses in seamount ecosystems. The results show that seamount sediments are reservoirs of extremely diverse and previously unknown viruses. Extensive interactions between viruses and dominant prokaryote lineages, as well as the presence of abundant AMGs in virus genomes, highlight the central roles viruses play in shaping the structure and function of seamount microbiomes and in influencing the biogeochemical processes mediated by seamount microorganisms. Furthermore, the geographical features of seamounts likely compromise the connectivity of viral communities, highlighting the important role of the topography of the deep-sea landscape in shaping local viral communities.

## Results and discussion

To explore the diversity, host interaction, and ecological function of viruses inhabiting deep-sea seamount sediments, 16S rRNA genes, metagenomes, and viromes were sequenced from seven sediment samples collected from the deep-sea seamount region in the Northwest Pacific Ocean. To fully consider the effect of the geographic features of seamounts on microbial and viral communities, sediment samples were collected across the seamount region. This region encompasses the C1 basin and three surrounding seamounts (i.e., NA, NLG, and MP4) with varying sampling locations in the bottom, hillside, and summit areas (Fig. 1 and Supplementary Data 1).

### Overview of prokaryotic communities

To assess the overall prokaryotic compositions in the sampled seamount sediments, the v3−v4 region of 16S rRNA genes was sequenced and analyzed. As shown in Supplementary Fig. 1, most prokaryotes in

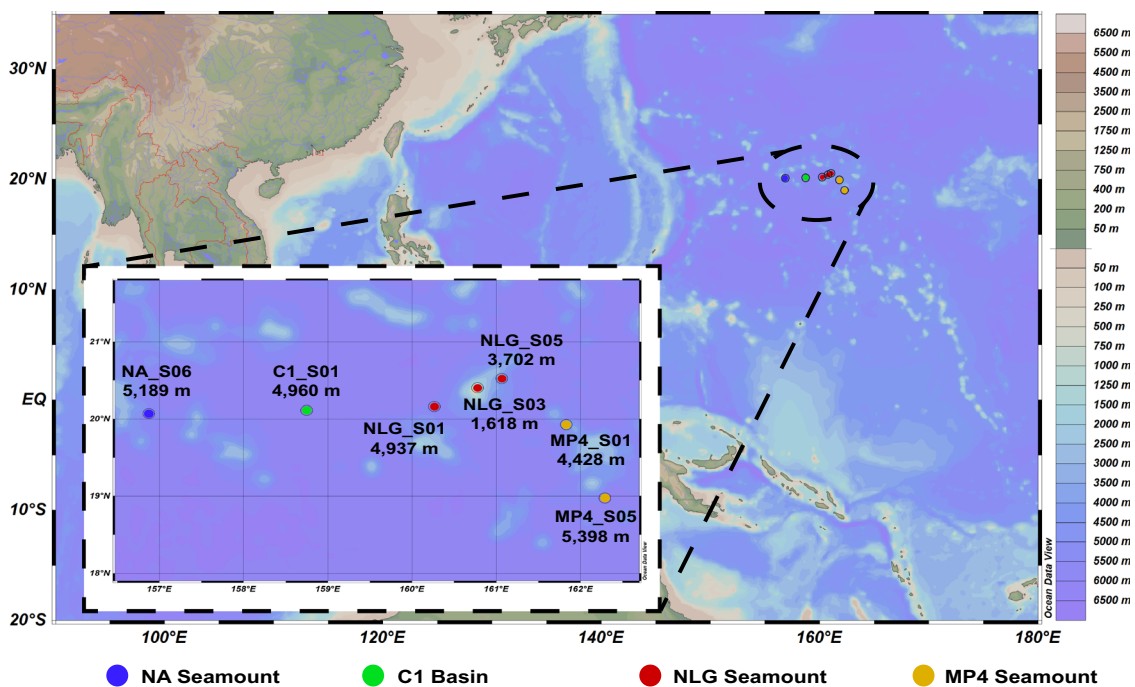

**Fig. 1 | Geographic distribution of sampling sites.** Sampling sites and depths are colored according to sampling environments. This image is plotted using Ocean Data View software (Schlitzer, Reiner, Ocean Data View, https://odv.awi.de, 2021). Source data are provided as a Source Data file.

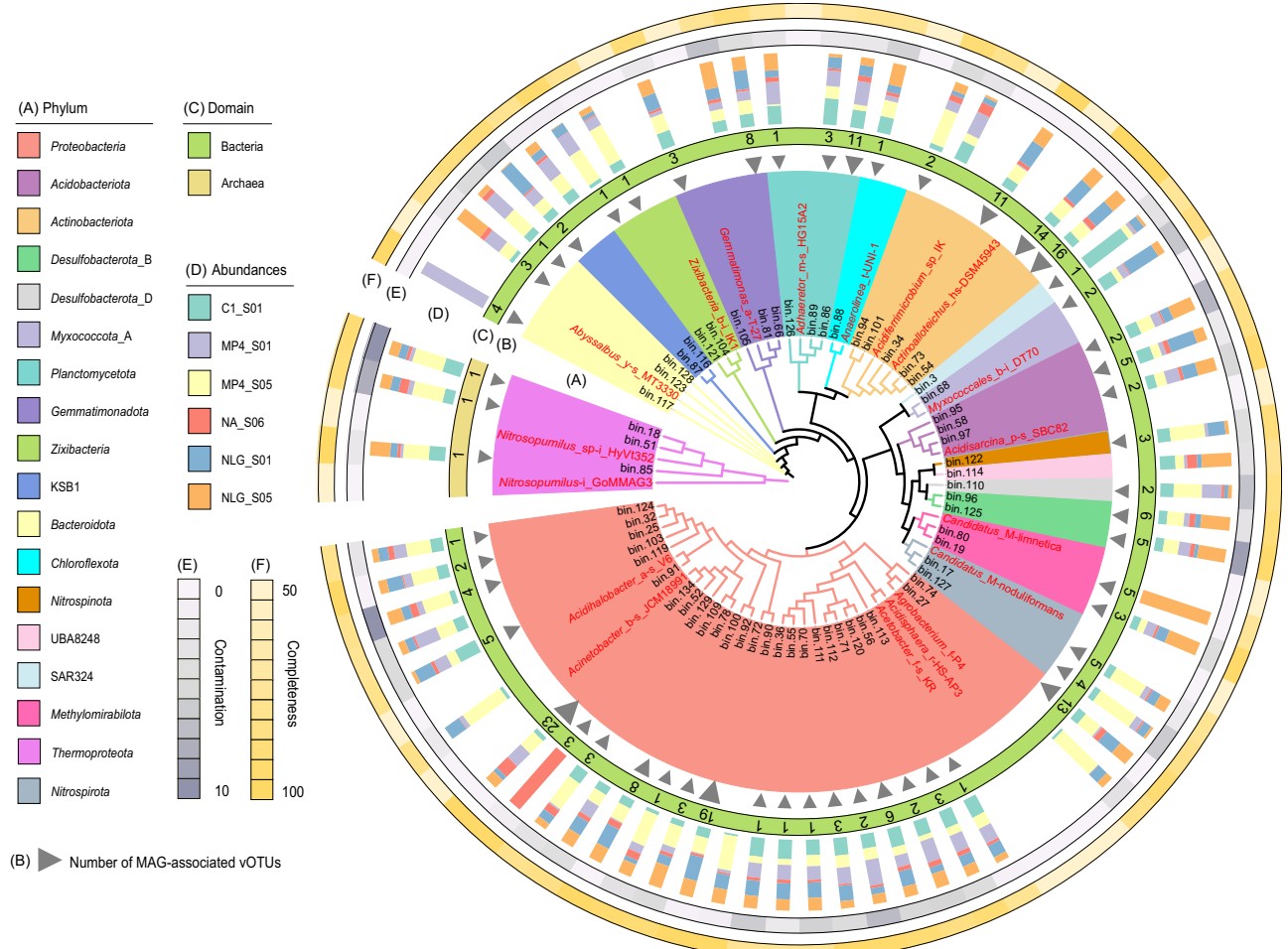

**Fig. 2 | Maximum-likelihood phylogenetic tree of prokaryotic metagenome-assembled genomes (MAGs).** The phylogenetic tree was inferred from the concatenated alignment of 120 bacterial or 122 archaeal single-copy marker genes. In the inter ring, clades are colored according to their annotated phylum, and the names of reference genomes and seamount sediment MAGs are presented in red and black text color, respectively. In the middle ring, MAGs from bacteria and archaea are in presented green or yellow, respectively. The numbers in the center ring and the size of triangles underneath indicate the number of seamount viral operational taxonomic units (vOTUs) predicted to infect corresponding MAGs. In the next outer ring, the stacked columns indicate the relative abundance of MAGs at different sampling sites. The outermost two rings show the completeness and contaminations of MAGs. Source data are provided as a Source Data file.

these sediments were bacteria, dominated by *Chloroflexi* (38.0% on average), *Proteobacteria* (35.7%), *Planctomycetota* (5.3%), SAR324 clade (Marine group B, 3.6%), *Actinobacteriota* (2.7%), *Acidobacteriota* (2.7%), *Nitrospinota* (1.2%), and *Firmicutes* (0.8%). *Chloroflexi* and *Proteobacteria* are the most abundant in all sediment samples, together accounting for more than 50% of the total relative abundance. Consistently, dominance of *Proteobacteria* in prokaryotic communities is also observed in a variety of deep-sea sediments, including cold seep, hydrothermal vents, and trenches[23–26]. Mantel's correlation analysis was performed to explore the effects of environmental physicochemical factors on prokaryotic communities, and the results showed that depth but organic carbon and nitrogen had a significant impact on prokaryotic communities (Supplementary Fig. 2).

De novo assembly and binning of metagenomes resulted in 136 high- or medium-quality microbial metagenome-assembled genomes (MAGs) with completeness ≥50% and contamination ≤10%[27]. These MAGs were then clustered at 95% average nucleotide identity (ANI) to generate 59 bacterial and three archaeal MAGs, representing species-level groups spanning 18 phyla (Fig. 2 and Supplementary Data 2). Most of the bacterial MAGs belong to dominant lineages, such as *Proteobacteria* (*n* = 26), *Actinobacteriota* (*n* = 5), *Acidobacteriota* (*n* = 3), *Planctomycetota* (*n* = 3), *Nitrospirota* (*n* = 2), *Chloroflexota* (*n* = 1), *Nitrospinota* (*n* = 1), and SAR324 (*n* = 1), while all archaeal MAGs belong to *Thermoproteota* (*n* = 3). These results were in line with the results of previous metagenome studies on microbial mat samples collected from Lōʻihi Seamount[28], and abyssal crust collected from Takuyo-Daigo Seamount[29]; in these samples, bacterial MAGs were dominated by *Proteobacteria* and archaeal MAGs accounted for less than 5% of the total MAGs. Based on the read coverage of MAGs among samples, while most of the MAGs were present in all six sediment samples, some of the MAGs were specific to certain samples; for example, bin.80 (*Methylomirabilota*), bin.52 (*Proteobacteria*), and bin.117 (*Bacteroidota*) were only present at sites NLG_S05, NA_S06, and MP4_S01, respectively (Fig. 2). In addition to 151 medium- and high-quality MAGs assembled from the metagenome of Axial Seamount and Lōʻihi Seamount samples[30–32], as well as the publicly available microbial genomes, these MAGs provide a good basis to infer linkages between viruses and prokaryotes.

## Viral community of seamount sediments

Currently, two metagenome approaches are available to study environmental viral communities, i.e., viromes of separated ambient viruses and bulk metagenomes containing sequences of diverse origins. Both approaches have powerfully expanded our knowledge of environmental viruses[33–35]. However, they differ greatly in their recovering efficiency of different viral populations, since viromes are greatly

enriched in ambient viruses whereas bulk metagenomes are depleted from certain free viruses and enriched in actively infecting and temperate viruses (cellular fraction)[36,37]. Therefore, in this paper, we utilize both bulk metagenome and virome to offer complementary perspectives of viral communities in seamount sediments. Three pipelines were used to identify viral sequences from bulk metagenome and virome datasets (Supplementary Fig. 3a), resulting in 2099 putative viral sequences (Contigs ≥5 kb or ≥2 kb and circular). Small circular Contigs (≥2 kb) were retained because they may represent small circular rep encoding single-stranded virus genomes of 2–25 kb size, such as phages belonging to *Microviridae* and eukaryotic viruses belonging to *Circoviridae* and *Geminiviridae*[38]. To obtain large viral assemblies, viral sequences were further binned using vRhyme v1.1.0[39]; the resulting viral Contigs and MAGs were then clustered at 95% identity and 85% coverage to generate 1600 viral operational taxonomic units (vOTUs) that represent approximately species-level taxonomy[40] (Supplementary Data 3). The GC content and size of these vOTUs ranged from 30.21 to 71.08% and from 2001 to 3,472,510 bp, respectively (Supplementary Fig. 3b). Fifty-nine vOTUs have a length of more than 200 Kb and possibly corresponded to giant viruses. Interestingly, three vOTUs are larger than 2.5 Mb, which is the approximately largest genome size of known isolated *Pandoravirus* viruses, a group of nucleocytoplasmic large DNA virus (NCLDV) related viruses that infect Ameba[41,42]. Further inspection of these three vOTUs showed that they all contain NCLDV marker genes, and a large fraction of their ORFs is homologous to NCLDV genomes (Supplementary Fig. 4). In addition, no known host contamination was found in these vOTUs, suggesting that they are likely bona fide NCLDV genomes. The high occurrence of extremely large viral genomes in our study may be due to the fact that we used Metaviralspades v3.15.5[43] for large contig assembly and vRhyme v1.1.0[39] for binning. Assessment of the quality of these vOTUs by CheckV v0.9.0[44] showed that 326 vOTUs (20.3%) were of medium quality and above, including complete (1.3%), high-quality (9.3%), and medium-quality (9.7%) (Supplementary Data 3 and Supplementary Fig. 3c). Mantel's correlation analysis showed that none of the three measured physicochemical factors (depth, organic carbon, and nitrogen) had a significant impact on viral communities (Supplementary Fig. 2).

Taxonomic affiliations of 1600 vOTUs were determined by comparing predicted ORFs against the NCBI viral_Refseq database (v94) based on the Last Common Ancestor algorithm. As shown in Fig. 3a and Supplementary Data 3, 88% of vOTUs (n = 1413) could be taxonomically affiliated at the phylum level, with primary assignment to *Uroviricota* (n = 1298, dsDNA tailed prokaryotic virus), *Nucleocytoviricota* (n = 60, dsDNA NCLDV), and *Artverviricota* (n = 31, reverse transcriptase-encoding ssRNA or dsDNA virus). Among *Uroviricota*, a total of 21 families were identified for vOTUs, including *Peduoviridae* (n = 36), *Kyanoviridae* (n = 21), *Autographiviridae* (n = 14), and *Mesyanzhinovviridae* (n = 12) (Fig. 3b). In all *Nucleocytoviricota*-affiliated vOTUs, matches associated with the families *Phycodnaviridae* (n = 26) and *Mimiviridae* (n = 7) were the most common (Fig. 3d), whereas viruses belonging to *Artverviricota* are most affiliated with the family *Metaviridae* (n = 26) (Fig. 3c).

To examine the viral community structures in seamount sediments, clean reads of virome and bulk metagenome were separately mapped to vOTUs to calculate reads per kilobase per million mapped reads (RPKM) values for each vOTU. As shown in Fig. 3e, although both virome and bulk metagenome identify abundant dsDNA viruses, ssDNA viruses and RNA viruses, they differ in the relative abundance of virus sub-populations. For example, except for *Duneviridae* and *Tectivirida*, nearly all detected dsDNA viral families showed higher relative abundance in the metagenome than in the virome. We suggest that this bias is likely caused by the filtration step used to remove hosts in the preparation of the virome, which may also remove certain large dsDNA viruses, in particular giant viruses belonging to NCLDV. In addition, a

portion of these dsDNA viruses may be temperate and thus difficult to be detected by viromes. Moreover, virome and metagenome have varying effectiveness levels for detecting ssDNA viruses depending on the target viral groups. For example, *Inoviridae* and *Circoviridae* were only detected in the virome and metagenome, respectively. The divergence of virome and bulk metagenome in the detection of different virus sub-populations was also observed previously in a comparative study of human gut virome and bulk metagenome. In that study, Gregory et al. found no significant difference between the bulk metagenome and virome in terms of the number of viral Contigs recovered, but the sub-population of viruses captured by the two approaches clearly differed, and the metagenome outperformed the virome in terms of the viral detection rate[36].

## Seamount viruses are diverse and novel

In seven seamount sediment samples, a significant portion of vOTUs were classified as members of the *Caudoviricetes* class, constituting ~81% of the total identified vOTUs. *Caudoviricetes* form a class of tailed dsDNA phages and are usually the most retrieved dsDNA viruses in the environment[33-35]. It is worth noting that the viral sequence databases still have very limited diversity represented, and the predicted dominance of *Caudoviricetes* in viromes is likely due to the high proportion of *Caudoviricetes* references in current databases. To infer DNA packaging mechanisms of these *Caudoviricetes*, a phylogenetic tree was generated based on a conserved gene coding for the terminase large subunit (TerL), which is necessary for DNA packaging during the maturation of tailed phages[45]. A total of 414 complete open reading frames (ORFs) encoding TerL were identified from seamount vOTUs and used for phylogenetic analysis along with the reference sequence (Supplementary Data 4). As shown in Supplementary Fig. 5a, seamount *Caudoviricetes* adopt diverse DNA packaging mechanisms, highlighting a remarkable diversity of *Caudoviricetes* within seamount sediment ecosystems.

ssDNA viruses are among the smallest and simplest viruses with genome sizes ranging from 2 to 25 kb. With the application of metagenomics and viromics, ssDNA viral sequences have been found to be abundant and diverse across various habitats[46], including marine sediments[47-49]. However, in our study, only a small fraction of vOTUs (n = 16) were identified as ssDNA viruses, including *Microviridae* (n = 11), *Circoviridae* (n = 2), *Inoviridae* (n = 1), *Spiraviridae* (n = 1), and *Smacoviridae* (n = 1). For *Microviridae*, eight complete or near-complete genomes (4361–5527 bp) were recovered from the seamount dataset. Blastn results showed that, according to NCBI's NT database, all of the seamount microviruses shared <93% identities relative to their best matches, highlighting the novelty of this viral family in seamount sediments. To obtain further insights into the evolution of seamount microviruses, their genome structures and gene sequence conservation levels were compared with known *Microviridae* genomes (Supplementary Fig. 5b). Like other known *Microviridae* genomes[50], the characteristic genes encoding well-conserved major capsid protein (VP1), DNA pilot protein (VP2), and replication protein (VP4) were identified in the genomes of all seamount microviruses. Six seamount microviruses that are homologous to *Gokushovirinae* possess an additional gene coding for internal scaffolding protein (VP3) and, in most cases, also a DNA binding protein (VP5). Nevertheless, certain seamount microviruses are quite different from known *Microviridae* genomes in terms of their gene organization and the content of unconserved genes. For example, the viral genome homologous to group D viruses (genome V_C1_S01_k141_1198016) displays distinct gene organizations from other microviruses and contains a gene downstream of *vp4* that exhibits no similarity with other group D viruses.

To further examine the diversity of viral communities in seamount sediments and their relationships with viral sequences identified from other marine habitats, a gene-sharing network was constructed using

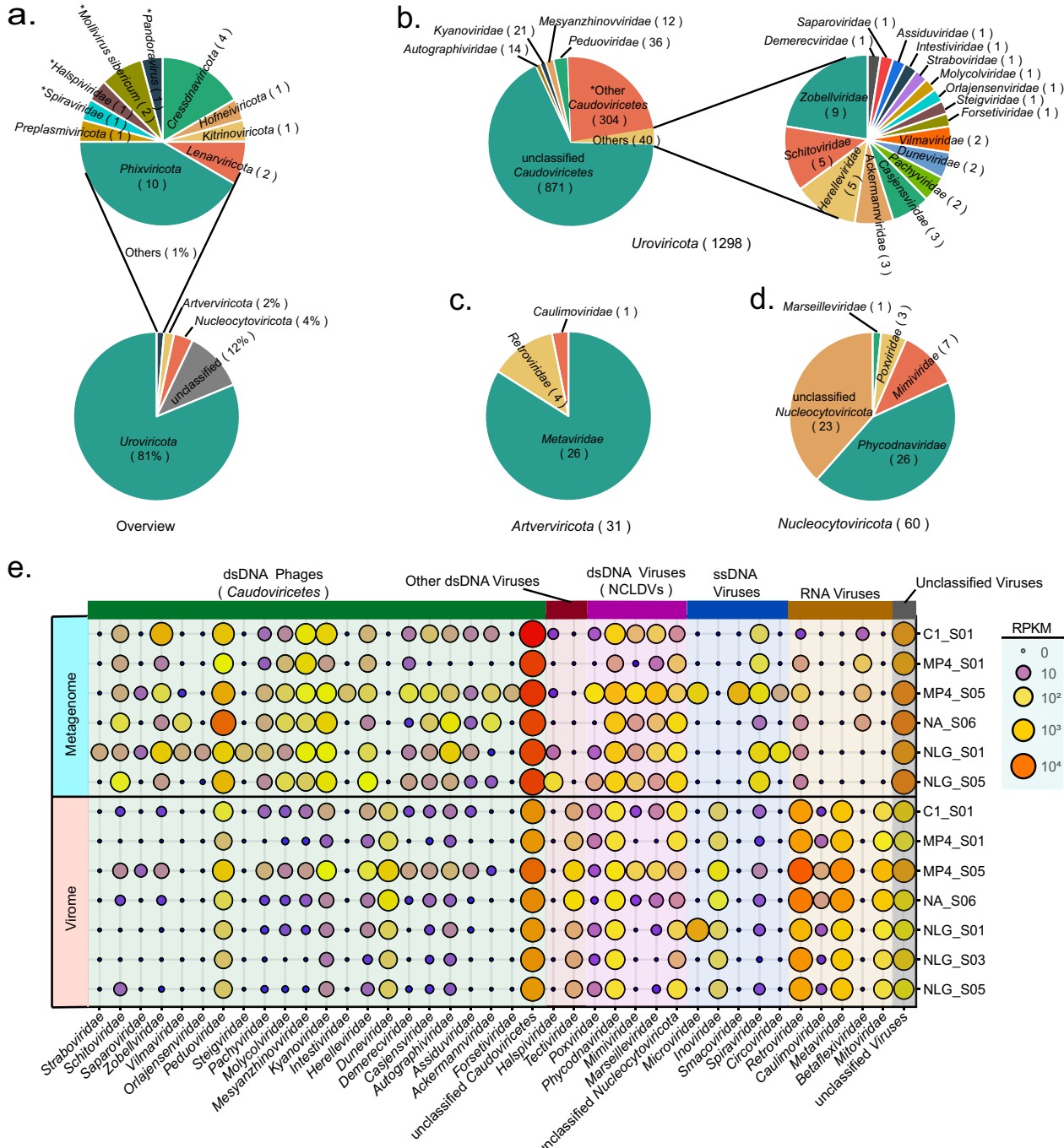

**Fig. 3 | Community structure of seamount sediment viruses. a** Relative percentage of viral operational taxonomic units (vOTUs) at the phylum level. vOTUs marked with asterisks have no taxon assignments at the phylum level and are therefore represented by the highest available taxon level. **b** The relative percentage of *Uroviricota*-affiliated vOTUs at the family level. The asterisk indicates vOTUs that have no taxonomic assignment at the family level. **c** Relative percentage of *Artverviricota*-affiliated vOTUs at the family level. **d** Relative percentage of *Nucleocytoviricota*-affiliated vOTUs at the family level. **e** Relative abundance of viruses in each seamount sediment sample at the family level obtained from metagenome or virome data. Source data are provided as a Source Data file.

vConTACT2[51]. Such a weighted network can assign viral sequences into viral clusters (VCs) that correspond to genus-level groups[51]. The vOTUs identified from GOV 2.0[52], cold seep[34], trench[53], and seamount were clustered into 17,518 VCs (Fig. 4a), while taxonomically known viruses from NCBI RefSeq only formed 345 VCs; this vast difference highlights the enormous and as yet undescribed diversity of marine viruses (Fig. 4a). As the largest marine virus database to date, GOV 2.0 datasets contributed the largest number of VCs ($n = 14,159$), while trench, cold seep, and seamount databases contributed 2171, 801, and 186 VCs,

respectively (Fig. 4a and Supplementary Data 5 and 6). In line with previous research[34,35], only five VCs were shared among all marine ecosystems, reflecting a high degree of variations in viral communities across various marine habitats (Fig. 4b). Among the 1600 vOTUs from seamount sediments, only 353 vOTUs were clustered into 186 VCs, the majority of which (77.94%) have no homologs in other marine databases and NCBI RefSeq databases; this suggests that most seamount sediment viruses are unique to the seamount habitat (Supplementary Data 6). Among the 186 seamount sediment VCs, only 18 VCs were

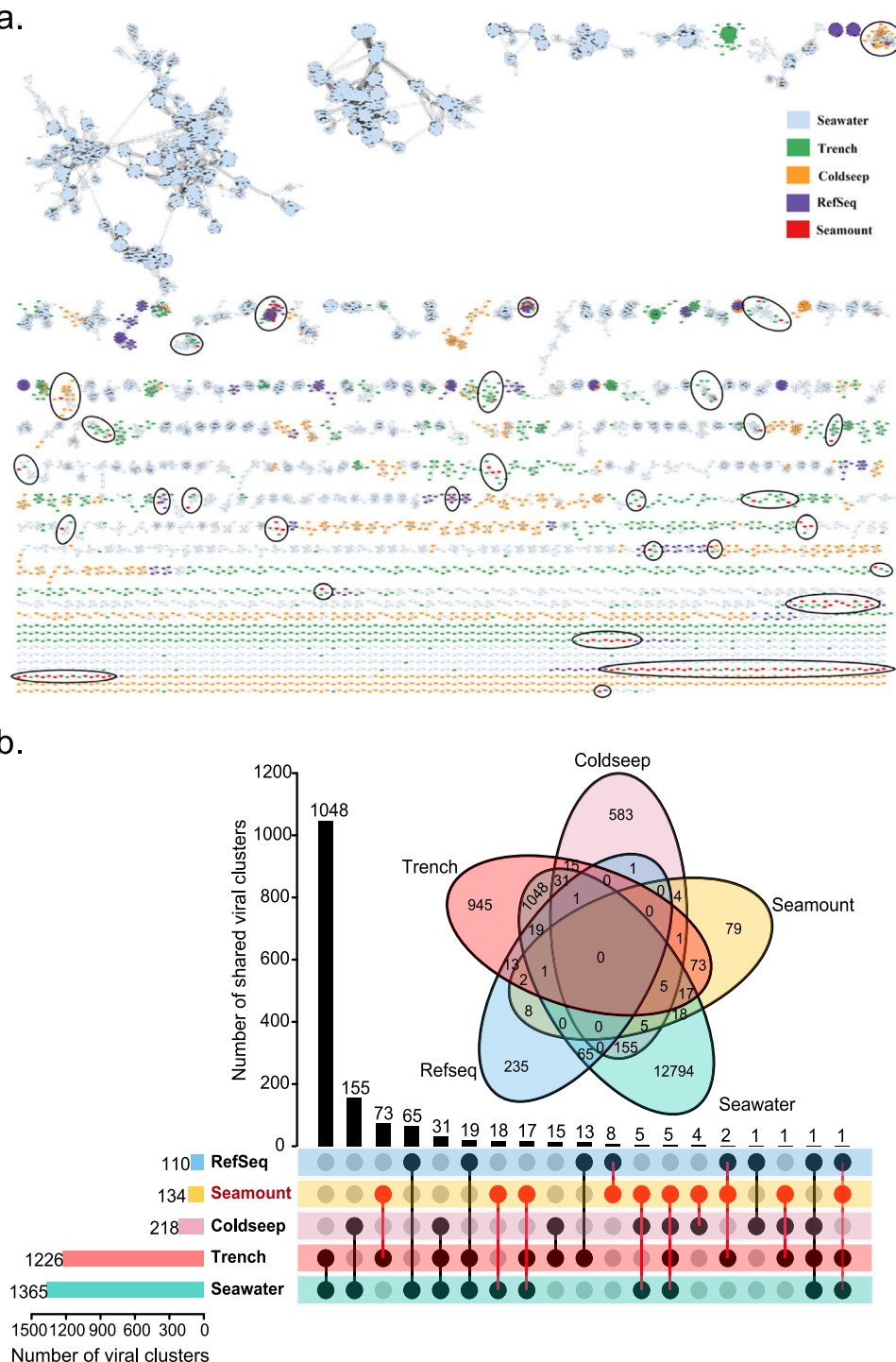

**Fig. 4 | Comparative analysis of seamount sediment viruses with RefSeq viruses and other viruses found in marine environments. a** Gene-sharing network of viral sequences from seamount sediment, seawater, trench, cold seep data sets, and RefSeq database. Viruses (nodes) are connected by edges, indicating the significant pairwise similarity between them in terms of shared protein contents. The positions of seamount viruses are marked with black circles. **b** Venn and upsets plots showing shared viral clusters among the environmental virus data sets and RefSeq database. Source data are provided as a Source Data file.

shared with the GOV 2.0 database, 73 VCs were shared with the trench database, 4 VCs were shared with the cold seep database, and 8 VCs were shared with the NCBI RefSeq database. The remaining 79 VCs (~42%) of viruses were exclusive to seamount viruses, which may represent candidate novel genera (Fig. 4b). These seamount-specific VCs contained 193 vOTUs, with the majority ($n = 161$, ~83%) affiliated as *Caudoviricetes*, followed by *Nucleocytoviricota* ($n = 16$), *Artverviricota* ($n = 5$), and ssDNA viruses ($n = 2$). The remaining nine vOTUs could not

be taxonomically assigned at the family or even higher level at this time.

## Virus-host linkages and viral lifestyles

Viruses affect various microbe-mediated processes through interactions with their hosts[13,54,55]. Considering that a large fraction of microbes is infected by viruses at any given time[56], the interactions between viruses and their hosts must play important roles in the

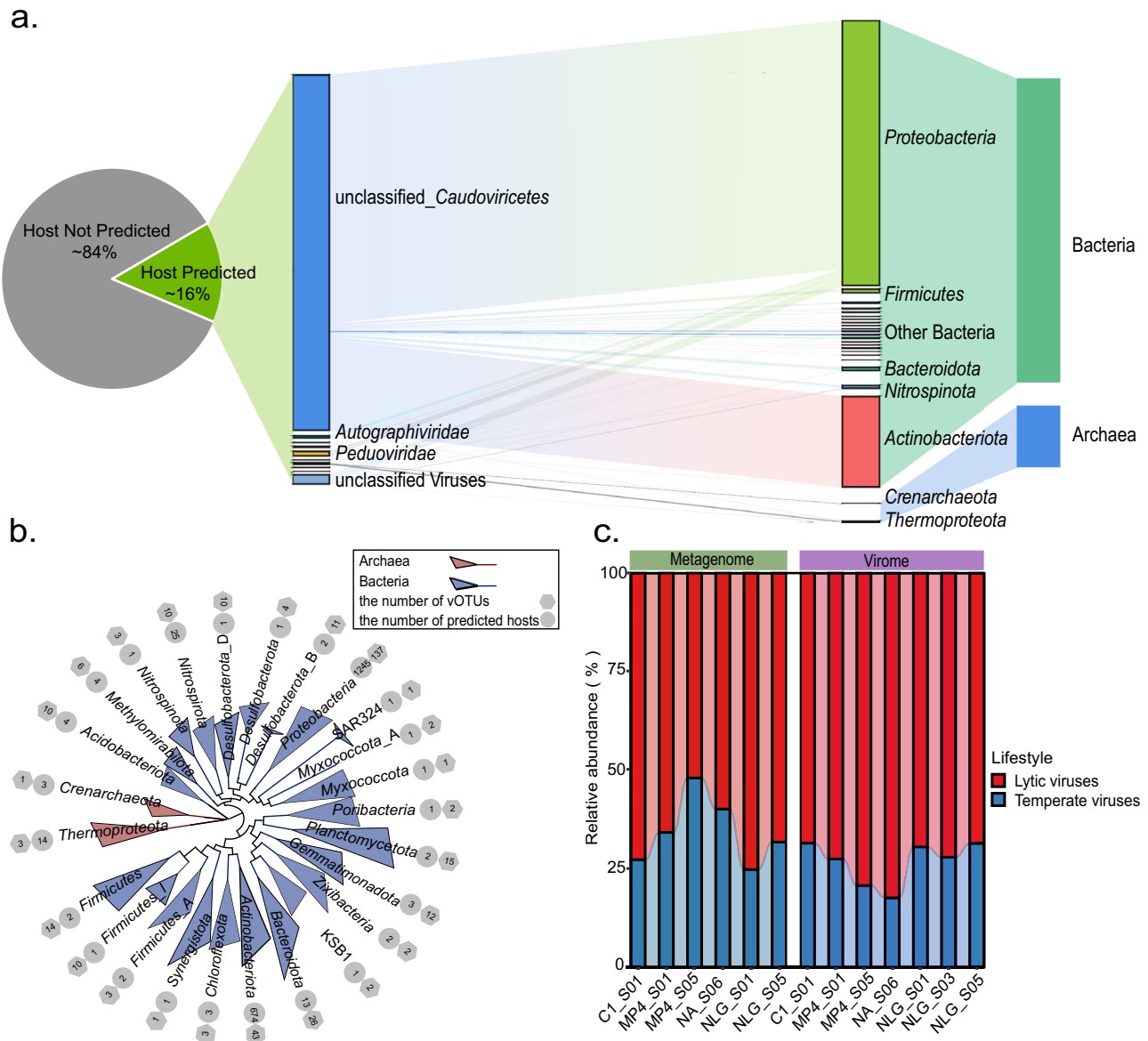

**Fig. 5 | Predicted host-virus interactions. a** Predicted virus-host linkages. Percentage and taxonomy of viral operational taxonomic units (vOTUs) for which a host was predicted are shown on the left; the taxonomy of predicted hosts is shown on the right. **b** Maximum-likelihood phylogenetic tree of predicted hosts. The phylogenetic tree was inferred from the concatenated alignment of 120 bacterial or 122 archaeal single-copy marker genes. The gray hexagons indicate the number of vOTUs predicted to have a host in the clade, while the gray circles indicate the number of hosts predicted to be infected by vOTUs in the clade. **c** Predicted life lifestyles for vOTUs obtained from either metagenome or virome. Source data are provided as a Source Data file.

dynamics, evolution, and ecology of microbial communities. To explore virus-host interactions in seamount sediments, potential hosts were predicted for 1600 vOTUs using a combination of four bioinformatic approaches, including CRISPR-spacers matching, tRNA matching, nucleotide sequence homology, and k-mer frequencies[57]. To predict virus-host connections, we used a combination of host databases to infer virus-host linkages, including the Genome Taxonomy database (GTDB-tk), MAGs binned from seamount samples in this study, and seamount samples from previous studies[30–32] (Supplementary Data 7). As a result, 3923 virus-host linkages were predicted, most of which were predicted by tRNA-matches ($n = 3316$), followed by nucleotide sequence homology ($n = 587$), CRISPR-spacers matches ($n = 54$), and k-mer frequencies ($n = 53$) (Fig. 5a and Supplementary Data 8). Among them, 75 virus-host connections were supported by two or more prediction approaches. Consistent with previous studies[34,35], putative hosts were predicted for only a small fraction

($n = 253$, ~16%) of the 1600 seamount vOTUs. Most of these vOTUs were predicted to infect specific hosts within the same phyla, and only 51 vOTUs were linked to a broader range of hosts across different phyla. These results agree with previous observations showing that most viruses only infect a narrow range of hosts[11,34,35]. A total of 2007 prokaryotes were predicted to be potential hosts for seamount vOTUs, most of which ($n = 1953$, ~97%) were predicted from the GTDB-tk database. Interestingly, all remaining potential hosts were predicted from the MAGs assembled from our study, and no hosts were predicted from MAGs assembled from other seamount metagenomes. This result implies the potential high divergence of viral communities across different seamount habitats.

Phylogenetic analysis showed that the predicted prokaryotic hosts of seamount viruses spanned two archaeal and 23 bacterial phyla (Fig. 5b). A total of 251 vOTUs were associated with bacteria. Of these, *Proteobacteria* was the most frequently predicted host phylum (137

associated vOTUs), followed by *Actinobacteriota* (43 vOTUs), *Bacteroidota* (26 vOTUs), *Planctomycetota* (15 vOTUs), and *Gemmatimonadota* (12 vOTUs). Most of these predicted hosts were among the most abundant bacterial lineages in the sampled seamount sediments, as indicated by the results of 16S rRNA gene profiling (Supplementary Fig. 1). For example, *Proteobacteria*—the dominant taxa in seamounts—contributed most (*n* = 2763, 70.4%) of the virus-host linkages. These results are in line with the widely recognized kill-the-winner hypothesis, which suggests that abundant microbes are more likely to be infected and lysed by viruses because a high population density increases the host-virus encounter rate[58]. Further taxonomic analysis showed that the most frequently predicted hosts within *Proteobacteria* were the subgroup *Gammaproteobacteria* (69 associated vOTUs) and *Alphaproteobacteria* (33 vOTUs); both subgroups are widely distributed across marine ecosystems, typically showing high abundance in seamount sediments and other deep-sea sediments[59–61]. Twenty-one *Proteobacteria* MAGs were predicted as hosts for 99 vOTUs (Fig. 2 and Supplementary Data 8). Remarkably, according to the host metabolic capability predicted based on the presence of metabolic genes within vOTU-associated MAGs, putative virus-infecting *Alphaproteobacteria* and *Gammaproteobacteria* may play important roles in carbon, nitrogen, and sulfur cycles in seamount sediments (Supplementary Fig. 6 and Supplementary Data 9). For example, hexosaminidase-encoding gene, a gene involved in chitin degradation, is widespread in vOTU-associated *Alphaproteobacteria* and *Gammaproteobacteria* MAGs, implying potential roles of these MAGs in complex carbon degradation. In addition, several virus-infecting MAGs contain *napA* and *nirBD* genes, which are involved in the reductions of nitrate to nitrite and nitrite to ammonia, respectively. This finding is in accordance with previous studies showing that *Proteobacteria* play an important role in connecting nitrifying and heterotrophic microorganisms, as they can reduce nitrate to ammonia and thereby provide a nitrogen source for other microorganisms[59,62]. Finally, previous studies suggest that *Gammaproteobacteria*, an important sulfur-oxidizing and sulfuric-acid-reducing taxon, provide an important contribution to sulfur biogeochemical cycling by being involved in and even driving sulfur transformations in sediments[60,61,63]. Indeed, abundant *dsrAB/sdo* genes were identified in vOTU-associated *Proteobacteria* MAGs; these genes are involved in the oxidation of hydrosulphides and are important for both the detoxification and neutralization of hydrogen sulfide in sediments (Supplementary Fig. 6). Collectively, given that *Proteobacteria* are abundant, highly active, and frequently associated with viruses, their infections and lyses by viruses likely substantially impact the microbial community and biogeochemical cycling in seamount sediments.

As another abundant and ubiquitous group inhabiting marine sediments[64], *Actinobacteriota* is the second most frequently predicted host phylum in seamount sediment samples and formed 946 virus-host linkages with 43 vOTUs. Four *Actinobacteriota* MAGs were predicted as hosts for 40 vOTUs (Fig. 2). Functional annotation of these four MAGs showed that they contain abundant genes involved in complex carbohydrate degradation, such as genes encoding for hexosaminidase, beta-glucuronidase, and isoamylase, which participate in the degradation of chitin, hemicellulose, and amylum, respectively (Supplementary Fig. 6 and Supplementary Data 9). Such complex carbohydrates are major components of crustacean shells, plant cell walls, and intercellular spaces, and they are also very difficult to degrade[65]; however, their biolysis is essential for biomass recycling in deep-sea sediments and critical in local and global carbon cycles. Therefore, infections and lyses of *Actinobacteriota* by viruses might play important roles in complex carbohydrate biolysis and carbon cycling.

Only four vOTUs were linked to archaea, including members of *Thermoproteota* (three vOTUs) and *Crenarchaeota* (one vOTU). The predicted archaeal hosts of *Thermoproteota* include three seamount

MAGs assembled in this study (Fig. 2), which form three novel virus–host linkages (not found in the IMG/VR V3 database) with two vOTUs, including a vOTU from *Caudovirales* and another vOTU that could not be taxonomically assigned at this time. Based on their taxonomical annotation, these three MAGs were further affiliated with the genus DRGT01 of the phylum *Thermoproteota*. In the GTDB-tk database, only seven MAGs are affiliated with archaea of DRGT01, all of which are derived from sediment samples, indicating that this group may be endemic to sediments and its connection with viruses has not been disclosed so far. As a ubiquitous group of archaea inhabiting various sediments, *Thermoproteota* is thought to be relevant to primary production in sediments through chemoenergetic autotrophic interactions and the ammonia oxidative metabolism[66–68]. Indeed, all of these three vOTU-linked seamount MAGs encode genes involved in the 3-hydroxypropionic acid/4-hydroxybutyric acid cycle that drive energy-efficient carbon fixation (Supplementary Fig. 6 and Supplementary Data 9)[69].

The lifestyles for 1600 vOTUs were predicted based on lysogeny-specific features, i.e., the presence of lysogeny-specific genes (e.g., genes encoding for integrase, recombinase, and excisionase) and/or location within their host genomes. As a result, approximately one-third (*n* = 548) of vOTUs were predicted to be lysogenic, 203 of which were predicted by both features. Based on abundances determined by read mapping, the relative abundance of temperate viruses in each sample was calculated for both virome and bulk metagenome datasets. As shown in Fig. 5c, both virome and bulk metagenome showed high occurrences of temperate viruses, accounting for averages of 34% and 27% of relative abundance in viral communities, respectively. As expected, the bulk metagenome exhibited a higher average ratio of temperate viruses than the virome, as it is conventionally enriched for genomes of temperate viruses that integrate into host genomes[36,37]. Given that a large fraction of viral genes is still poorly annotated at this time, certain bona fide lysogeny-specific genes may be neglected. Moreover, the incomplete assembly of viral genomes also makes it difficult to detect temperate viruses by identifying either lysogeny-specific genes or flanking host sequences, which further leads to an underestimation of lysogeny signals. Thus, temperate viruses may be even more prevalent in seamount sediments than identified in this study. This is further supported by the fact that 84% of all MAGs assembled here can be linked to vOTUs by nucleotide sequence homology matches; consequently, these may represent sequences that are acquired by the host through phage genome integrations[57]. The prevalence of temperate viruses has also been observed in a variety of deep-sea environments, such as deep water from the South China Sea and the western Pacific Ocean[70] as well as deep-sea diffuse-flow hydrothermal vents[71]. Previous studies have suggested that the high occurrence of temperate viruses in the deep sea possibly promotes virus-mediated gene transfer and exchange, which may be important for the survival and stability of hosts in challenging environments[70–72].

## Potential impacts of viral AMGs on host metabolisms and biogeochemical cycles

Viruses can reshape the metabolism of their hosts through the expression of virus-encoded AMGs[73,74]. To better understand the impacts of viral AMGs on host metabolisms and relevant biogeochemical cycling, AMGs were identified from vOTUs by VIBRANT and DRAMv pipelines; they were further functionally annotated with Pfam, KEGG, and CAZy databases. As a result, after manual curation, a total of 331 genes were identified as putative AMGs (Supplementary Data 10). Because viruses generally obtain and maintain AMGs from their hosts, we further performed sequence homology searches of AMGs against the NCBI NR database to predict their putative hosts. As shown in Supplementary Fig. 7a, a large proportion (*n* = 112, 33.8%) of these AMGs were probably derived from *Proteobacteria*, which is consistent

with *Proteobacteria* being the most frequently predicted hosts for vOTUs in seamount sediments (Fig. 5a).

Based on KEGG annotation, the putative AMGs of seamount viruses were involved in diverse metabolic pathways, with a large portion participating in carbohydrate metabolism, cofactors and vitamins metabolism, as well as amino acid metabolism (Supplementary Fig. 7b). Notably, several AMGs were involved in carbon, nitrogen, and sulfur cycling. For example, six AMGs were affiliated with glycoside hydrolases predicted to catalyze the hydrolysis of complex polysaccharides, including trehalase and amylase. In marine sediments, these genes are essential for the recycling of detrital organic matter supplied from the overlying water column and thus may be critical in local and global carbon cycles[75]. Several AMGs were associated with sulfur cycling, including genes encoding for phosphoadenosine phosphosulphate reductase (CysH), cysteine synthase (CysK), sulfate adenylyltransferase (Sat), and methanethiol oxidase (SELENBP1). CysH and CysK participate in assimilatory sulfate reduction, whereas the Sat is involved in dissimilatory sulfur reduction/oxidation; both reactions are important for sulfur cycling[76]. SELENBP1 catalyzes the oxidation of methanethiol, which is a significant step in the sulfur cycle as methanethiol is an intermediate of the metabolism of globally important organosulfur compounds, including dimethylsulphoniopropionate[77]. The most common AMG related to nitrogen cycling is nitronate monooxygenase (*ncd2*), which catalyzes the oxidation of nitroalkane to nitrite[78]. Seamount microorganisms are involved in carbon degradation and sulfur, nitrogen, and metal cycling[9,79]. Therefore, the presence of these AMGs indicates that seamount viruses may extensively participate in local and global biogeochemical cycles by assisting microbes in driving biogeochemical cycles with AMGs.

To explore the relative abundance of AMGs, clean reads of both the virome and bulk metagenome were separately mapped to AMG-carrying vOTUs. As shown in Fig. 6a, virome and bulk metagenome showed different AMG profiles. In general, these results indicated a significantly elevated relative abundance of AMGs associated with processes such as protein folding, sorting, and degradation glycan biosynthesis and metabolism, xenobiotics biodegradation and metabolism, as well as lipid metabolism within the virome in comparison to bulk metagenome. Conversely, AMGs related to energy metabolism exhibited diminished abundance within the virome when compared with the metagenome. In addition, the bulk metagenome exhibited greater AMG diversity in pathways related to carbohydrate metabolism, amino acid metabolism, and metabolism of cofactors and vitamins. This clear dissimilarity in AMG compositions and the abundance between the virome and the bulk metagenome is likely caused by their bias in enrichment for viruses of different lifestyles (Fig. 5c). Luo et al. found that the viral lifestyle was more important than habitat and prokaryotic host in driving viral AMG profiles[74]. They suggested that lytic viruses tended to encode AMGs that could boost progeny reproduction, whereas temperate viruses tended to encode AMGs for host survivability. Consistently, similar trends were also observed in our study. For example, the virome that was more enriched for lytic viruses exhibited higher relative abundances of *fabG* and *queC-E*; *fabG* is involved in fatty acid synthesis, while *queC-E* plays a role in redirecting host protein synthesis, thus improving host translation efficiency and viral progeny reproduction[80]. In contrast, glycoside hydrolases were more diverse and frequently encoded in the bulk metagenome, which was more enriched for temperate viruses. Such AMGs potentially facilitate the decomposition and utilization of complex carbohydrates in hosts in deep-sea sediments, thereby enhancing host adaptation to their environments.

To gain deeper insights into the functions of seamount AMGs, the specific impacts of AMGs on the host metabolism were further examined based on predicted host-virus linkages. As shown in Fig. 6b, 16 AMG-carrying vOTUs were predicted to infect 13 seamount MAGs (Supplementary Data 11 and 12), forming 16 host-virus linkages. All of these vOTUs are tailed phages belonging to *Caudoviricetes*, while the hosts spanned seven phyla, including *Bacteroidota*, *Proteobacteria*, and *Nitrospirota*. Interestingly, all these vOTUs are predicted to be temperate viruses, which tend to encode AMGs that benefit both hosts and viruses[74]. Indeed, among 10 host-virus linkages, viral AMGs exert potential compensatory effects on the host metabolism because their homologs were not found in host MAGs. These AMGs were involved in a variety of host metabolisms, including energy metabolism (*cysH* and *SELENBP1*), metabolism of cofactors and vitamins (*ubiG*, *nadM*, and *hspA*), amino acid metabolism (*tyr*, *dnmt1*, and *phnZ*), glycan biosynthesis and metabolism (*kdsA*, *nadM*, and *hspA*), and carbohydrate metabolism (*aceB*). For instance, in the sulfate reduction step of the sulfur cycle pathway, the MAG of bin.117 encodes two genes (*cysC* and *cysN*) that catalyze the conversion of sulfate to phosphoadenylyl sulfate (PAPS) but lack the *cysH* gene that catalyzes the reduction of PAPS to sulfite (Fig. 6c). However, its associated vOTU (V_MP4_S05_vRhyme_bin_185) contains AMGs encoding for CysH. Therefore, lysogenic viral infection likely compensates for host metabolic capabilities in sulfate reduction, further highlighting the potentially important roles of viral AMGs in sulfur cycling. In the other six virus-host linkages, host MAGs contain the homologs of AMGs that are carried in associated vOTUs. The impacts of these viral AMGs on host metabolisms are unknown to date, but we suspect that they may augment host metabolic flux by overexpressing host metabolic genes[81].

## The impacts of seamount geographical features on local viral communities

Several studies on fauna have suggested that isolation of seamount habitats would promote localized speciation, giving rise to high levels of endemism on seamounts[1,82]. This so-called "seamount endemism hypothesis" was challenged by accumulating morphological and genetic evidence on the fauna, suggesting that seamounts do not generally support high levels of endemism[83,84]. To examine whether geographic features of seamounts cause high divergence in viral communities, we used two levels of information: (1) viral genetic space represented by protein clusters (PCs)[85] and (2) species-level viral populations represented by vOTUs. Totals of 790,756 PCs and 1600 vOTUs were identified across the seven samples. The accumulation curves based on pan PCs (Fig. 7a above) and vOTUs (Fig. 7a below) showed that the 5th samples and beyond still added PCs and vOTUs, but also depicted a trend to approach a plateau. These results suggest that, although it is impossible to obtain a complete sample, viral genetics and populations (in particular dsDNA viruses) from seamount sediments are relatively well sampled. Comparative analysis showed that 66% (n = 522,201) of PCs were only present within a single sample, and only 3.3% (n = 25,932) of PCs were shared by all seven samples, suggesting a high divergence of viral genetic space in these samples (Fig. 7b above). Similarly, comparative analysis of vOTUs on the population level showed that most vOTUs (n = 1526, 95.4%) were unique to a single sample, further supporting the remarkably high level of divergence among viruses across seamount sediments (Fig. 7b below).

To further explore whether the geographical features of seamounts contribute to the high divergence of seamount viruses, the connectivity of viral populations between neighboring seamount sampling sites was calculated. As shown in Fig. 7c, neighboring sites not separated by seamounts generally showed strong connectivity among viral populations, whereas neighboring sites across the same seamount (i.e., with the summit in the middle) showed decreased connectivity. For example, NLG_S03 and NLG_S05 sites, which are located on two different seamounts but have no large geographic barrier between them, showed high correlation in their viral communities. However, the MP4_S01 site showed considerably weaker connection with the MP4_S05 site, even though both are located on the same seamount but on opposite slopes. Moreover, regardless of the substantial difference in the depth between MP4_S03 and

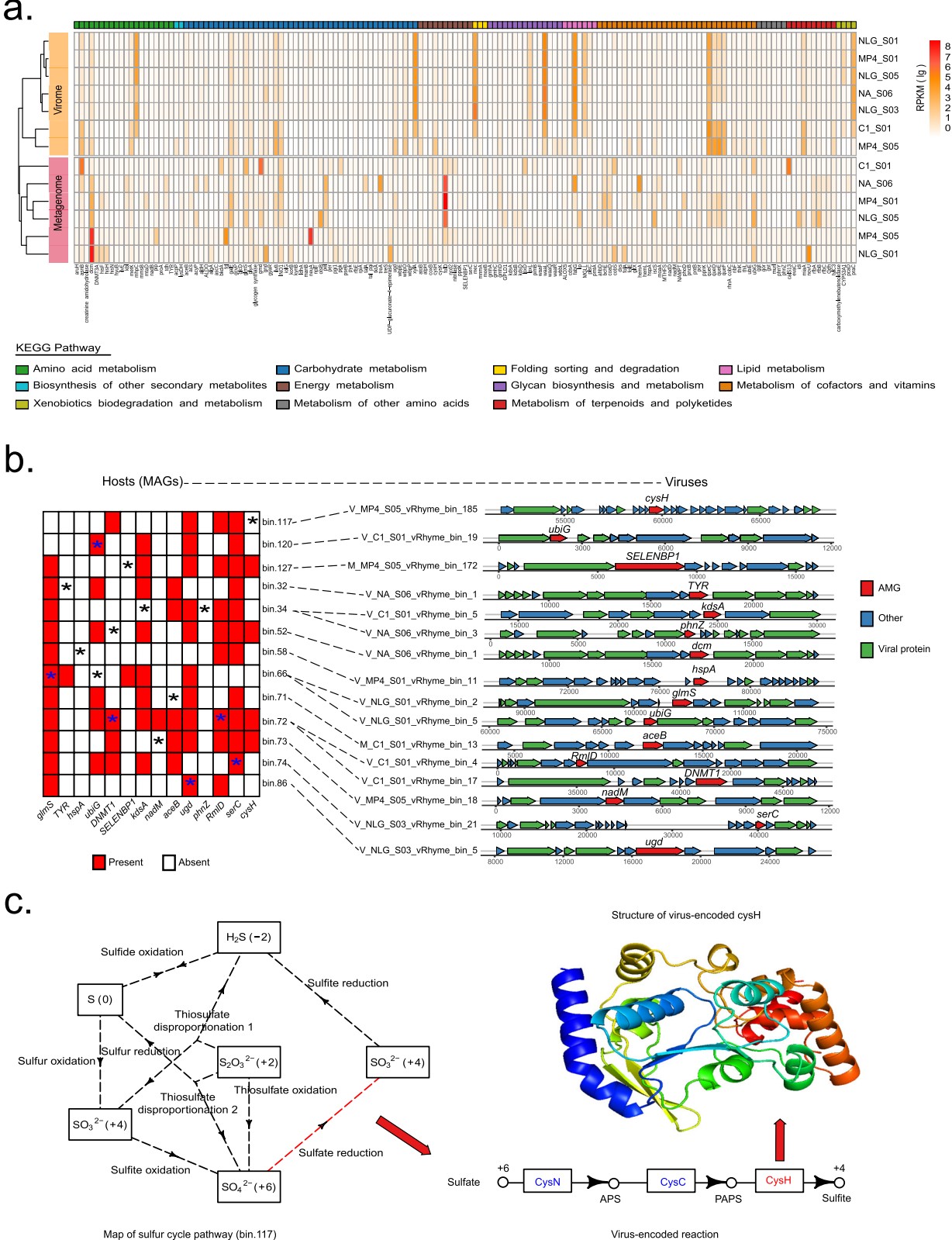

**Fig. 6 | Effects of viral auxiliary metabolic genes (AMGs) on host metabolism.**
**a** Relative abundance of viral AMGs obtained from virome or metagenome dataset in different seamount sediment samples. Kyoto Encyclopedia of Genes and Genomes (KEGG) metabolic categories are colored according to the legend.
**b** Metabolic linkage of metagenome-assembled genomes (MAG) and viral AMGs. Dashed lines indicate the inferred host-virus infection relationships. For MAG (left),

the presence and absence of the homologs of viral AMGs in the MAG are indicated in red and white, respectively. The asterisk indicates the AMG that is carried by the corresponding virus. For viruses (right), the genomic context of AMG (red) is shown. **c** The compensation of host sulfur cycle (bin.117) by the virus-encoded cysH gene. Viral AMG and virus-encoded reactions are marked in red. Source data are provided as a Source Data file.

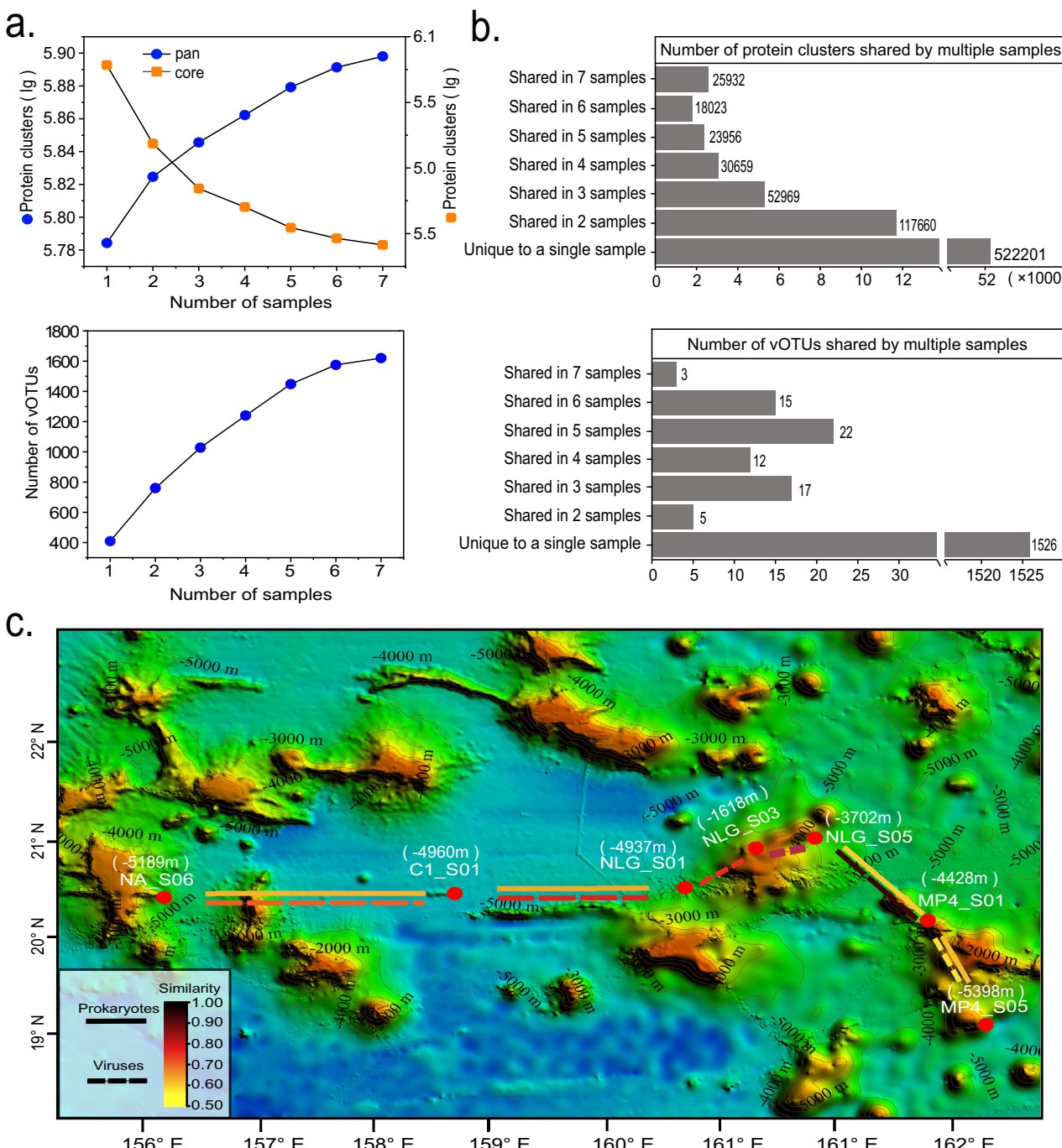

**Fig. 7 | The effects of geographical features of seamounts on viral community.** **a** Accumulation curves of viral protein clusters (PCs, above) and viral operational taxonomic units (vOTUs, below) in the seamount dataset. **b** Number of viral PCs (above) and vOTUs (below) shared by multiple samples. **c** Calculated connectivity of viral (dashed line) and prokaryotic (solid line) populations between neighboring seamount sampling sites. The connectivity of viral and prokaryotic populations was calculated based on the similarity in the prokaryotic operational taxonomic unit (pOTU) and vOTU profiles, respectively. The gradient scale indicates the similarity between neighboring seamount sampling sites. Source data are provided as a Source Data file.

NLG_S05 sites, both displayed much stronger connectivity than most pairs of neighboring sites, even though some pairs have comparable depths (such as C1_S01 and NLG_S01). This observation suggests that depth is not one of the primary factors causing divergence in viral communities across seamount sites, which was also supported by the result of Pearson correlation analysis ($p > 0.05$). Collectively, our results suggest that the physical barrier of the seamount rather than the isolation of the seamount or site depth may explain the degree to which viruses are highly divergent across seamounts. Because viruses require host organisms to replicate, we further attempted to examine the impact of seamount geographical features on the connectivity of prokaryotes. Although the connectivity of the prokaryotic community was also generally impaired by seamounts, the prokaryotic community did not follow the same connectivity pattern that was observed for the viral community (Fig. 7c). The different geographic distribution patterns between viral and prokaryotic communities may be explained by the fact that the factors influencing prokaryotic communities are complex and diverse[86,87]. While the virus is indeed a significant factor in

shaping host communities, other factors such as grazing and environmental conditions (e.g., depth, salinity, temperature, and nutrients) also play important roles in shaping prokaryotic communities[86,87]. In addition, although our results showed that virus and host communities in deep-sea seamount sediments are interacting through close associations, a substantial proportion of seamount viruses was revealed to be temperate (Fig. 5c), which tend to coexist with hosts, thus normally don't directly impact host communities by causing host mortality. Previous studies have suggested that a significant fraction of sediment viruses are derived from sinking within the overlying seawater, either on host cells or on particulate matter[88,89]; some of these viruses may even persist in marine sediments for more than thousands of years[88]. The viral population could be passively transported on oceanic currents[90] and seamounts exert complex effects on deep ocean circulation and mixing (e.g., deflection of major currents)[3]; therefore, we suggest that the physical barrier of the seamount suppresses the passive dispersal of ambient viruses by comprising the mixing effects of deep currents, thereby leading to a high level of divergence in viral communities. This study provides the first evidence on the effects of seamount geographic features on the assembly of local viral diversity; however, closer integration of molecular, oceanographic, geological, and ecological research with more well-characterized sediment samples is needed in the future to verify our hypothesis.

## Methods

### Sample collection and physicochemical measurements

Deep-sea seamount sediment samples were collected from the seamount region in the Northwest Pacific (Fig. 1 and Supplementary Data 1) during the COMRA cruise DY45 in July and August 2017. To fully consider the effect of the geographic features of seamounts on microbial and viral communities, sediment samples were collected across seamounts with varying locations in the bottom, hillside, and summit areas (Fig. 1 and Supplementary Data 1). NA sample was collected from the base of the NA Seamount, and C1_S01 sample was collected from the sediment of the C1 basin between the NA and NLG Seamounts. NLG_S01, NLG_S03, and NLG_S05 samples were collected from the base, summit, and hillside of the NLG Seamount, respectively. MP4_S01 and MP4_S05 samples were collected from the base of the MP4 Seamount but on opposite sides. The sediment samples were collected using a multi-tube sampling approach, and only sediment layers at 2–6 cm below the sea floor were selected for further analysis. The detailed geological information on collected sediments is listed in Supplementary Data 1. The collected sediment samples were frozen at −80 °C on board until further analysis. Nutrient concentrations, including total nitrogen and total organic carbon, were determined at the Qingdao Science Standard Testing platform (Qingdao, China) using standard methods.

### Taxonomic profiling of microbial communities by 16S rRNA gene analysis

The total DNA was extracted from sediment samples using the Powersoil DNA Isolation Kit (Mo Bio, USA) according to the manufacturer's instructions. The V3–V4 region of the 16S rRNA gene was amplified from total DNA using the forward primer 341F (5′ CCTACGGGNGGCWGCAG3′) and the reverse primer 805R (5′GACTACHVGGGTATCTAATCC3′). The library was prepared using the TruSeqTM DNA Sample Prep Kit (Illumina, USA) and sequenced on the Illumina MiSeq platform (Illumina Inc., San Diego, CA, USA) by Majorbio Bio-Pharm Technology Co., Ltd. (Shanghai, China). The raw data was first imported into the QIIME 2 v2022.2.0 pipeline[91] via the "tools import" command to produce an a.qza format file suitable for downstream analysis. Sequence quality was assessed via the demux plugin, followed by quality control, denoising, and generation of OTU tables using the DADA2 plugin. Taxonomic annotation was performed using the SILVA database (132_99_16S) via the feature-classifier plugin.

### Bulk metagenomic sequencing and assembly

For metagenomics sequencing, a paired-end library was generated from total DNA using NEXTFLEX Rapid DNA-Seq Kit (Bioo Scientific, USA) and sequenced on an Illumina NovaSeq 6000 platform (Illumina Inc., San Diego, CA, USA) by Majorbio Bio-Pharm Technology Co., Ltd. (Shanghai, China). The raw reads were trimmed and quality filtered using fastp v0.23.2[92] to generate clean reads with high quality. The Contigs were then assembled from clean reads using MEGAHIT v1.2.9[93] software (--k-list 21, 29, 39, 59, 79, 99, 119, and 141), and subsequently quality assessed using QUAST v5.2.0[94].

### Generation and analysis of prokaryotic metagenome-assembled genomes

The Contigs assembled from the metagenome were binned by the MetaWRAP v1.3.0[95] binning module based on maxbin2, metabat1, and metabat2 methods. The original bins were refined using the MetaWRAP v1.3.0[95] bin_refinement module (with parameters -c 50 -m 10), which were then quality checked by CheckM v1.0.12[96]. The high- and medium-quality bins (completeness ≥50% and contamination ≤10%) were then aggregated and dereplicated at 95% ANI using dRrep v3.3.0[97], resulting in a total of 63 species-level MAGs. MAGs were taxonomically assigned using GTDB-tk v2.1.0[98] based on classify_wf workflows. Maximum-likelihood phylogeny of MAGs was inferred using IQ-TREE v2.2.0.3[99] from a concatenation of 120 bacterial or 122 archaeal marker genes produced by GTDB-tk v2.1.0[98]; the generated tree was visualized using iTOL v4 (https://itol.embl.de/). To determine the relative abundance of MAGs in each sample, clean reads were mapped to MAGs using CoverM v0.6.1[100] (with parameters -contig -m rpkm --trim-min 5 - -trim-max 95) to calculate RPKM values. Functional annotation of MAGs was performed using METABOLIC v4.0[101] (-m-cutoff 0.75 -kofam-db full) (Supplementary Data 9).

### Virus purification

Ambient viruses were purified from seamount sediment samples according to methods described previously[33]. Briefly, 20 g of seamount sediment sample was suspended in 30 ml of SM solution (100 mM NaCl, 8 mM $MgSO_4 \cdot 7H_2O$, and 50 mM Tris/HCl; pH 7.5), shaken for 30 min at 4 °C and centrifuged at 3000 × g for 15 min at 4 °C. The precipitated sediment particles were then repeatedly extracted with SM solution, and the resulting supernatants from both extractions were combined. After filtering through a 0.45 μm mesh, the viral particles in the supernatant were enriched using 100 kDa centrifugal ultrafiltration tubes by centrifugation at 4000 × g until the final sample volume measured less than 1 ml. We used 0.45 μm filters to enrich ambient viruses instead of 0.22 μm filters, because 0.45-μm filtration offers advantages such as avoidance of missing large viruses crucial for assessing diversity, more comprehensive virus detection, and reduction of processing losses[102].

### Virome sequencing and assembly

Prior to viral DNA extraction, virus concentrates were treated with DNase I at 37 °C for 1 h to remove exogenous DNA. Encapsidated viral DNA was then extracted as described by Thurber et al.[103]. The library was prepared using the TruSeq DNA Sample Prep Kit (Illumina, USA) and sequenced on the Illumina HiSeq 2000 platform (Illumina Inc., San Diego, CA, USA) by Majorbio Bio-Pharm Technology Co., Ltd. (Shanghai, China). The raw reads were trimmed and quality filtered using fastp v0.23.2[92] to generate clean reads with high quality. Contigs were then assembled from clean reads using MEGAHIT v1.2.9[93] software (--k-list 21, 29, 39, 59, 79, 99, 119, 141).

### Identification of viral sequences

To obtain as many bona fide viral sequences as possible, viral sequences were identified by the following three pipelines: (i) Contigs ≥2 kb from metagenome assemblies, as well as virome assemblies,

were used to recover viral sequences by VirSorter2 v2.2.3[104] and VIBRANT v1.2.0[105] using default settings. CheckV 0.9.0[44] was then used to evaluate the quality of viral sequences and remove host contaminations. Only viral sequences containing at least one of the viral hallmark genes (such as virion morphogenesis gene and terminase gene) were retained. (ii) What the Phage (Wtp) v1.1.0[106] was used to identify viral sequences from virome assemblies and only complete, high-quality, or circular viral sequences recognized by CheckV v0.9.0[44] were retained. (iii) Metaviral SPAdes v3.15.5[43] was used to assemble and identify viral sequences from the clean reads of the virome, and only complete, high-quality, or circular viral sequences recognized by CheckV v0.9.0[44] were retained. Viral sequences identified by pipelines (i) and (ii) were combined and binned using vRhyme v1.1.0[39] to generate vMAGs. The remaining un-binned sequences were combined with viral sequences identified by pipeline (iii) and were regarded as vContigs. The resulting vContigs/vMAGs were de-redundant by vRhyme v1.1.0[39] (with parameters --derep_only --method longest --derep_id 0.95 --frac 0.80) to generate vOTUs. An overview of the bioinformatic workflow used for the identification of viral sequences is shown in Supplementary Fig. 3a.

## Taxonomic assignments and abundance profiles of viruses
ORFs were predicted from vOTU sequences by Prodigal v2.6.3[107] (-p meta -g 11 -m -c). Predicted ORFs were then mapped against the NCBI viral_Refseq database (2023-04-26) using CAT v5.0.3[108] to determine the taxonomic affiliation of vOTUs based on the Last Common Ancestor algorithm. To determine the relative abundance of vOTUs in each sample, clean reads were mapped to vOTUs using the contig mode and genome mode of CoverM v0.6.1[100] (with parameters -m rpkm --trim-min 5 - -trim-max 95), to calculate RPKM values.

## Mantel's correlation analysis
Pairwise comparisons of environmental variables (organic carbon, depth, and nitrogen) were calculated by Pearson correlation analysis. The correlations between Bray–Curtis dissimilarity of prokaryotic and viral OTU profiles and Euclidean distances of environmental variables were assessed using the Mantel test with the "ggcor" package in R[109,110]. Each environmental variable was related to prokaryotic and viral OTU profiles by the Mantel test (function mantel; permutations = 9999 and method = "pearson"). Finally, Pearson correlations and the Mantel test were visualized using R software via the "ggcor" package[109,110].

## Prediction of virus-host linkages
Virus-host linkages were predicted based on four different in silico strategies[35]: (1) CRISPR-spacers match. A CRISPR spacer database was constructed for a set of microbial genomes using MinCED v0.4.2[111]. CRISPR spacers were then queried for exact sequence matches against viral Contigs using BLAST+ v2.9.0[112] (with parameters -task blastn-short -word_size 7 -perc_identity 95 -qcov_hsp_ perc 95). Only matches of at least 95% identity over 95% spacer length and only ≤1 mismatch were regarded as highly confident virus-host linkages. (2) tRNA match. tRNAs were identified from microbial genome dataset and vOTUs using ARAGORN v1.2.41[113]. Recovered tRNAs were matched using BLASTn v2.9.0[112] (blastn -perc_identity 100 -qcov_hsp_perc 100), and only those with exact match (100% identity over 100% coverage) were regarded as highly confident virus-host linkages. (3) Nucleotide sequence homology. Sequences of vOTUs were compared with the dataset of microbial genomes by BLASTn v2.9.0[112] (blastn -perc_identity 70 -qcov_hsp_perc 75 -e$^{-3}$). The match requirements were 70% minimum nucleotide identity, 75% minimum coverage of the viral contig length, 50 minimum bit score, and 0.001 maximum e-value. (4) k-mer frequencies. WIsH v1.0[114] was run using default parameters to infer a connection between viruses and hosts based on k-mer frequencies, and $p \le 0.005$ was considered as a match (Supplementary Data 8). Whenever multiple hosts were predicted for a specific vOTU,

the one supported by multiple approaches was chosen. The microbial genome dataset used above was composed of (1) all reference genomes from GTDB-tk (release207_v2, $n = 24{,}778$), (2) All MAGs (≥50% completeness and ≤10% contamination, $n = 62$) recovered from seamount metagenomes in this study, and (3) other seamount MAGs obtained from IMG/MR ($n = 151$, Supplementary Data 7).

## Life strategy prediction
Life strategies of vOTUs were predicted by the following two pipelines. (1) VIBRANT v1.2.0[105] and CheckV v0.9.0[44] were used to infer temperate life strategies by identifying vOTUs that contain provirus integration sites or integrase genes. (2) ORFs from all vOTUs were functionally annotated using eggNOG-mapper v2.1.6[115], and sequences containing lysogeny-specific genes (i.e., genes encoding for integrases, recombinases, transposases, excisionases, CI/Cro repressor, and parAB) were selected and manually inspected. The vOTUs identified by the above pipelines were considered temperate, while others were considered unknown.

## Construction of phylogenetic trees on major virus groups
Phylogenetic trees of *Caudovirales* and *Microviridae* were generated based on TerL and VP1, respectively. The Contigs assembled from the seamount sediment metagenome and the virome were searched by HMMER v3.3.2[116] (hmmsearch model) using the Hidden Markov Model (PF03237.hmm, PF04466.hmm, and PF05876.hmm for TerL, and PF02305.hmm fro VP1); sequences with an e-value ≤ 0.05 were retained. The obtained sequences and reference sequences (Supplementary Data 4 and 13) were aligned by MUSCLE v5.1[117]. The alignments were then trimmed using TrimAL v1.4.rev15[118] (-automated1). The maximum-likelihood phylogenetic tree was then constructed using IQ-TREE v2.2.0.3[99] (-bb 1000 -nt AUTO -m MFP), and the support for nodes in the trees was evaluated with 1000 bootstrap replicates. The resulting phylogenetic trees were visualized by iTOL (https://itol.embl.de/).

## AMG identification and annotation
Viral AMGs were identified and annotated using both VIBRANT v1.2.0[105] and DRAMv v1.3.5[119] pipelines as follows: (1) VIBRANT pipeline. The AMGs in vOTU sequences were identified by VIBRANT v1.2.0[105] using default parameters. (2) DRAMv pipeline. The VirSorter2 v2.2.3[104] (--prep-for-dramv) was run on the vOTU sequences, and the AMGs (auxiliary scores <4) were predicted from the resulting sequences by DRAMv v1.3.5[119]. The AMGs predicted by the above pipelines were combined, and manual curation was carried out to remove illegal AMGs (such as nucleotide metabolism, DNA-related reactions, modification of viral components, modification of viral components, ribosomal proteins, transcriptional/translational regulators, and viral invasion). Putative AMGs were further annotated using the dbCAN2 server (https://bcb.unl.edu/dbCAN2/) and NCBI CD-search tool (https://www.ncbi.nlm.nih.gov/Structure/cdd/wrpsb.cgi) with the threshold value of e-value < 10$^{-5}$. To infer the host source of AMGs, they were queried against the NR database (2021-01-07) by BLASTp. For the prediction of the AMG 3D structure, Colabfold v1.5.2[120] was used (Supplementary Data 10 and 11). The relative abundance of AMGs is determined by calculating the relative abundance of vOTUs carrying those AMGs in each sample.

## Comparisons to viral sequences from other marine environments and RefSeq database
To compare the viral sequences from seamount sediments with those from other marine environments and the RefSeq database, protein-sharing network analysis of seamount sediment vOTUs, Global Oceans Viromes 2 (GOV 2.0) database[52] (>10 kb, $n = 195{,}728$), and viral Contigs from cold seep ($n = 2885$)[34] and trench ($n = 12{,}700$)[53] were performed. For each viral Contig, ORFs were predicted using Prodigal v2.6.3[107] (-p meta -g 11 -m -c), and predicted protein sequences were then subjected

to all-to-all BLASTp using Diamond v2.0.15[121]; the result file was used as input for vConTACT2 v0.11.3[51]. Viral RefSeq (v94) was used as reference database to generate the protein-sharing network (Supplementary Data 5 and 6), and Cytoscape v3.9.1[122] was used to visualize the network.

## The effects of geographical features of seamounts on viruses
To generate viral core and pan PCs in the seamount virus dataset, ORFs were called from virome-assembled Contigs using Prodigal v2.6.3[107] and were then aligned to the GOV2.0[52] and IMG/VR (release version 2022-12-19_7.1)[123] database using Diamond v2.0.15[121] with a threshold of e-value ≤ 1e$^{-5}$, identity ≥ 30%, and coverage ≥ 50%[53]. The resulting ORFs were translated into proteins and clustered at 60% identity and 80 coverage using CD-HIT v4.6[124] to generate non-redundant viral PCs. Pan PCs were obtained by merging PCs from different samples, and core PCs were acquired by identifying the PCs that were shared by all samples.

The connectivity of viral and prokaryotic populations between neighboring seamount sampling sites was calculated based on their similarity in the pOTU and vOTU profiles, respectively. Briefly, pOTU and vOTU matrix tables were used as import files for the vegan package of the R software[110,125]; the Bray–Curtis distance matrix between each sample was calculated and further converted to similarity values using the following formula: similarity value = 1/(1 + distance matrix)[126].

## Plotting
Box plots, heat maps, bar stacking plots, and gene maps were drawn using the R packages ggplot2 v4.3.2[127], pheatmap v1.0.12[128], ggpubr v0.6.0[129], and gggenes v0.5.0 (https://cran.r-project.org/web/packages/gggenes), respectively. Venn and upset plots were plotted by Tbtools v1.120[130].

## Reporting summary
Further information on research design is available in the Nature Portfolio Reporting Summary linked to this article.

## Data availability
The raw data of bulk metagenome, virome, and 16S rRNA genes are deposited in the NCBI SRA database under accession numbers SAMN36987747-52, SAMN36987740-46, and SAMN36987753-58, respectively. All processed data generated in this study are provided in Supplementary Data files (Supplementary Data 1–13). Source data for all main and Supplementary Figs. are provided with this paper. The raw Treefiles for phylogenetic trees in Fig. 2 (https://doi.org/10.6084/m9.figshare.25245436), Fig. 5b (https://doi.org/10.6084/m9.figshare.25245730), and Supplementary Fig. 5 (https://doi.org/10.6084/m9.figshare.25245745) are deposited in the Figshare repository. The raw data file for the gene-sharing network in Fig. 4a is deposited in the Figshare repository (https://doi.org/10.6084/m9.figshare.25245358). The links to the databases used in this study are listed below: Silva database (release 132) [https://www.arb-silva.de/documentation/release-132/]; Genome Taxonomy database [https://data.ace.uq.edu.au/public/gtdb/data/releases/release207/]; NCBI RefSeq database [https://ftp.ncbi.nlm.nih.gov/refseq/release/viral/]; NCBI Taxonomy database [https://www.ncbi.nlm.nih.gov/taxonomy]; eggNOG database (release 5.0) [http://eggnog5.embl.de/download/eggnog_5.0/]; Pfam [https://pfam.xfam.org/]; dbCAN2 server [https://bcb.unl.edu/dbCAN2/]; NCBI CD-search tool [https://www.ncbi.nlm.nih.gov/Structure/cdd/wrpsb.cgi]; Global Oceans Viromes 2 (GOV 2.0) database [https://datacommons.cyverse.org/browse/iplant/home/shared/iVirus/GOV2.0]; viral Contigs from cold seep [https://doi.org/10.6084/m9.figshare.12922229]; viral Contigs from trench (OEP001086 and OEP001087) [https://www.biosino.org/node/]; IMG/VR database (release 2022-12-19_7.1) [https://genome.jgi.doe.gov/portal/IMG_VR/

IMG_VR.home.html]; IMG/MR [https://img.jgi.doe.gov/]. Source data are provided with this paper.

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

## Acknowledgements

This work was financially supported by the Scientific Research Foundation of Third Institute of Oceanography, MNR (2024025, M.J.), the National Natural Science Foundation of China (41976084, M.J.), the Innovation Group Project of Southern Marine Science and Engineering Guangdong Laboratory (Zhuhai) (311021006, M.J.), and the Deep Sea Habitats Discovery Project (DY-XZ-04, M.J.).

## Author contributions

M.J., Z.S., and R.Y.Z. conceived the project. M.J. performed the generation of virome and bulk metagenome. M.Y., M.Z., M.J., and R.C. performed data analyses. M.Y. and M.J. wrote the manuscript with contributions from Z.S., R.Z., Y.H., F.K., X.F., X.D, and Y.L. All authors read and approved the manuscript.

## Competing interests

The authors declare no competing interests.
