## [Peer Review File · Nature Communications]

REVIEWER COMMENTS

Reviewer #1 (Remarks to the Author):

The manuscript "Globally novel and locally endemic viruses inhabiting deep-sea seamount sediments" presents a thorough and systematic study evaluating bacteria, archaea, and viruses in several seamount sediments. The authors use genetic information recovered from bacterial, archaeal, and viral communities to identify differences in the composition of these three communities from different, but relatively close, locations. The primary observations highlighted in the abstract suggest that there is evidence of extensive and novel diversity, extensive virus-host interactions, significant roles for viruses in carbon, sulfur and nitrogen cycling, prevalence of temperate phages with the potential to modify host metabolisms, and geographic limitations to virus connectivity. These are among the many characteristics of viral communities in nature and highlights their known roles in ecology and evolution. The manuscript is generally well written and this work provides a significant amount of new sequence information with corresponding analyses.

Broader comments

My key concern is that the manuscript is an excellent descriptive account but one that lacks a significant novel finding. For example, one of several key findings listed in the title, abstract, and conclusion is that viruses in seamount sediments are locally endemic. The introduction uses this concept as a key justification for the study (lines 132-144) and it is discussed in some detail in the last paragraph and conclusion, but the data suggesting that there is evidence of local endemism is not presented until the end of the paper (Figure 8). It seems unusual to have a novel or key finding discussed at the end of a manuscript. Much of what comes before is a very detailed and systematic description of microbial and viral diversity at different sites. To make this a more compelling argument, the manuscript needs some oceanographic or geological data that describe the types of sediment samples that were collected.

One of the other issues I have with the locally endemic finding is that data highlighting host-virus interactions and the explanation for virus endemism are somewhat contradictory. In the section that covers this topic, the authors note that viruses and prokaryotes do not exhibit the same patterns and suggest that endemism for viruses is likely due to geographic features that limit viral dispersion through mixing (lines 573-588). The authors bring this up again in the conclusion, but they also note that there was extensive interactions between viruses and dominant prokaryotic lineages, as well as the occurrence of AMG's with the potential to shape community structure and function (lines 596-600). If virus and host communities in sediments are interacting through close associations, how does dispersal through physical processes such as mixing impact one community and not the other? Doesn't the interaction data suggest that viruses and their hosts are connected in sediments? Wouldn't this imply that even if they had different sources, they have changed together over time and should therefore have similar trends regarding geographic distributions?

Along those lines, there is also no physical mixing data and very little chemical data. Nutrient measurements for carbon and nitrogen are mentioned in the methods and appear in one supplemental table, but with no discussion. The authors rule out depth of the water column as a driving factor, but there is no information about the method used to collect or characterize the sediments. How were the sediments selected, collected, characterized? Can the differences in sample types, chemistry, age, depth, etc. tell us anything about the types of sediments that were collected?

Lastly, the accumulation curves in Figure 8 (panel a) suggests that protein clusters and vOTUs captured most of the diversity in the samples. However, there is no description of how protein clusters were determined, only a reference. And viral identifications were limited using strict criteria to identify virus genes that were similar to viral genes in other known viruses. Both methods are prone to systematic errors, in that identifications of new protein clusters and viruses are limited to what is in the published search database.

More specific comments:

Lines 344-347: The authors state that 97% of all virus hosts (1,953) were predicted from MAGs in GTDB and that all remaining hosts were from seamount sediment MAGs generated in their study, with no MAGs identified as hosts from other sediment samples. Does this mean that only 60 sediment MAGs were potential hosts? This seems like a relatively small number to make the broad statement that this implies endemism across different seamount habitats. Aren't seamount habitats under sampled? The author's show data which suggest that very different virus communities are identified depending on the method used to sequence them, such as a metagenome versus a metavirome. Can samples be compared across different seamounts? If so, how many studies have been conducted? Do they have similar depth of coverage?

Much is missing from the methods:

1. Sediment selection, collection, and characterization details are missing
2. DNA extraction methods are missing.
3. Details about viral purification. Many bacteria will pass through a 0.45 micron mesh. Was bacterial DNA identified in the metaviromes? If so, how much of it was bacterial and how was it removed?

Reviewer #2 (Remarks to the Author):

This study uses metagenomic and viromic sequencing and bioinformatic efforts to explore how microbial and viral community diversity, virus-host relationships, and viral ecological roles differ between various seamounts. The authors find extensive viral diversity and identify several viral-host relationships. AMGs encoded by viruses may ultimately play a role in influencing host metabolism and affect carbon, nitrogen, and sulfur cycling in these sediments while also enabling host survival in these environments. Authors also reveal that no hosts were predicted from MAGs assembles from other seamount sediments suggesting endemism of viral communities within seamounts.

I thoroughly enjoyed reading this paper. I feel that this paper went above and beyond by analyzing prokaryotic and viral communities in seamount sediments, identifying viral diversity, virus-host linkages, and viral lifestyles, impacts of viruses on host metabolism, and searching for impacts of seamount features on the presence of local viral communities. In some cases, I think the analyses went beyond the current capabilities/knowledge base of the field and potentially made some bold statements that I feel can't be justified (i.e., classifying viral taxonomy based on a somewhat conserved viral gene). Overall, the work does support the conclusion and the claims. The methods are sound and thorough. I do believe this is the first paper assessing viral diversity and ecological roles in deep-sea seamount sediments, which is extremely exciting.

I have made some specific comments on areas where I would improve the paper, but ultimately these notes are more focused on removing or rephrasing information as opposed to adding additional analyses.

Specific comments:

The introduction holds a lot of great information, but the first two paragraphs could be made more concise.

It would be good to include citations for the statement made in Line 133.

Line 190-194: This seems a bit obvious. I would remove "as several studies have suggested." I think it should be reworded because these are two very different methods and viral recovery is not coincidentally better if you are enriching for viruses. Typically, metagenomes are acquired by filtering onto 0.2um filters and therefore viruses slip through and are not accounted for during sequencing (with the exception of temperate phage and viruses that are actively infecting the cellular fraction). I think this is made clear towards the end of the sentence, so I would remove "as several studies have suggested" because this really is common practice and knowledge.

Line 208: "to huge viruses"

I believe they are more commonly referred to as "giant viruses"?

Line 239:

Couldn't this also be because these viruses may be temperate or actively infecting the hosts (i.e., are not ambient and wouldn't otherwise be detected in the free viral population)?

Figure 3e: This is a great figure.

Line 254: Might be worth noting the bias that the viral database itself plays in recognizing Caudovirales viruses (since these makes up a majority of the viral database). It therefore makes sense that the viral database continues to largely identify Caudovirales.

Lines 261-267: I don't think that placing viral sequences in a phylogenetic tree and making inferences on lineages is important in the goal of this paper, and I don't think it is correct practice. (If it is, several papers need to be cited to support this method and claim.) It is complicated to assess viral phylogeny because they do not have conserved genes, and even if a gene is conserved, I'm not sure you can make claims regarding viral diversity based on this one gen. You also come to this conclusion/ show this in Lines 290-295.

Figure 4: It is very hard to tell which are the reference genes as the grey circles are too small in 4a. I would find a way to make this more obvious. I also feel that more justification is needed in order to try to classify different viruses based on a shared gene. Figure 4a doesn't seem to mean very much as it is now. Are you trying to define taxonomy based on a gene? If so this should be its own paper in itself. I would either remove Figure 4 or move it to the supplemental. I would also be more clear on what your goal is in trying to map viral phylogeny. If it is for viral diversity, then I would be careful not to make claims about viruses forming different groups based on a gene that really can't be used as an identifier since they are not conserved. Instead you can focus on Figure 4b and talk about how genome arrangements are different, but again this is expected, so consider moving this to the supplements.

Figure 5a: The text could explain what we are looking at a bit better. The caption may want to include the organization of nodes from greater to fewer.

Figure 5b: The two numbered scales to the left need labels to be more clear. I love this figure.

Line 330 & 343: The full name is GTDB-tk and should be used as such.

Line 346: Very interesting!

Line 607: I believe VLPs refer to viral like particles, not necessarily ambient or free viruses. I would describe these as ambient viruses instead of VLPs.

Overall, I would absolutely recommend this paper for publishing.

Responses to the reviewers' comments (point-to-point)

Reviewer 1:

Comment#1:The manuscript “ Globally novel and locally endemic viruses inhabiting deep-sea seamount sediments” presents a thorough and systematic study evaluating bacteria, archaea, and viruses in several seamount sediments. The authors use genetic information recovered from bacterial, archaeal, and viral communities to identify differences in the composition of these three communities from different, but relatively close, locations. The primary observations highlighted in the abstract suggest that there is evidence of extensive and novel diversity, extensive virus-host interactions, significant roles for viruses in carbon, sulfur and nitrogen cycling, prevalence of temperate phages with the potential to modify host metabolisms, and geographic limitations to virus connectivity. These are among the many characteristics of viral communities in nature and highlights their known roles in ecology and evolution. The manuscript is generally well written and this work provides a significant amount of new sequence information with corresponding analyses.

Response: Thanks very much for your positive feedbacks. Your comments have all been most valuable and helpful for improving the manuscript. The manuscript has been extensively revised according to your comments as listed below. All the changes in this revision are marked in Red color for easy recognition.

Comment#2: My key concern is that the manuscript is an excellent descriptive account but one that lacks a significant novel finding. For example, one of several key findings listed in the title, abstract, and conclusion is that viruses in seamount sediments are locally endemic. The introduction uses this concept as a key justification for the study (lines 132-144) and it is discussed in some detail in the last paragraph and conclusion, but the data suggesting that there is evidence of local endemism is not presented until the end of the paper (Figure 8). It seems unusual to have a novel or key finding discussed at the end of a manuscript. Much of what comes before is a very detailed and systematic description of microbial and viral diversity at different sites. To make this a more compelling argument, the manuscript needs some oceanographic or

geological data that describe the types of sediment samples that were collected.

Response: To the best of our knowledge, this is the first paper assessing viral communities in deep-sea seamount sediments in-depth, and many findings reported here are novel to viruses inhabiting seamount ecosystems. We provide detailed and systematic descriptions of viral diversity and virus-host interactions in the earlier parts of the manuscript, because these data are very important for understanding the diversity and ecological roles of seamount viruses, and are thus also one of the key findings. In addition, the investigation of local viral endemism is a more in-depth and environmentally specific analysis integrating seamount geographical features, which is built on our previous data on viral diversity and virus-host interaction in the earlier parts of the manuscript. Thus, we put the descriptions of local viral endemism at the end of the manuscript.

As for the response to the comment “the manuscript needs some oceanographic or geological data that describe the types of sediment samples that were collected”, please refer to the response to Comment#4 below.

Comment#3: One of the other issues I have with the locally endemic finding is that data highlighting host-virus interactions and the explanation for virus endemism are somewhat contradictory. In the section that covers this topic, the authors note that viruses and prokaryotes do not exhibit the same patterns and suggest that endemism for viruses is likely due to geographic features that limit viral dispersion through mixing (lines 573-588). The authors bring this up again in the conclusion, but they also note that there was extensive interactions between viruses and dominant prokaryotic lineages, as well as the occurrence of AMGs with the potential to shape community structure and function (lines 596-600). If virus and host communities in sediments are interacting through close associations, how does dispersal through physical processes such as mixing impact one community and not the other? Doesn't the interaction data suggest that viruses and their hosts are connected in sediments? Wouldn't this imply that even if they had different sources, they have changed together over time and should therefore have similar trends regarding geographic distributions?

Response: It is a very good question. The reasons for the seemingly “contradictory”

are as follows:

(i) In fact, the factors influencing prokaryotic communities are complex and diverse (Ambati and Kumar, 2022; Mojica and Brussaard, 2014). While the virus is indeed a significant factor in shaping host communities, other factors such as grazing and environmental conditions (e.g., depth, salinity, temperature, and nutrients) also play important roles in shaping prokaryotic communities (Ambati and Kumar, 2022; Mojica and Brussaard, 2014).

(ii) Although our results showed that virus and host communities in deep-sea seamount sediments are interacting through close associations, a substantial proportion of seamount viruses was revealed to be temperate (Fig. 5C), which tend to coexist with the host, thus don't shape host communities directly by causing host mortality.

To make the manuscript more clear, we have provided the above explanations for the different geographic distribution patterns between viral and prokaryotic communities in Lines 578-587.

References

Ambati M, Kumar M S. Microbial Diversity in the Indian Ocean Sediments: An Insight into the Distribution and Associated Factors [J]. *Current Microbiology*, 2022, 79(4): 115.

Mojica K D A, Brussaard C P D. Factors affecting virus dynamics and microbial host–virus interactions in marine environments[J]. *FEMS Microbiology Ecology*, 2014, 89(3): 495-515.

Comment#4: Along those lines, there is also no physical mixing data and very little chemical data. Nutrient measurements for carbon and nitrogen are mentioned in the methods and appear in one supplemental table, but with no discussion. The authors rule out depth of the water column as a driving factor, but there is no information about the method used to collect or characterize the sediments. How were the sediments selected, collected, characterized? Can the differences in sample types, chemistry, age, depth, etc. tell us anything about the types of sediments that were collected?

Response: We have linked the nutrient measurements with viral and prokaryotic communities using RDA analysis. The relevant data and descriptions are provided in

Supplementary Fig. 2 and Lines 155-158 and 214-215. The detailed information regarding the sediment selection, collection, and characterization has been provided in Lines 625-639. Besides, the detailed geological information on the types of sediments has been provided in Supplementary Table 1.

Comment#5: Lastly, the accumulation curves in Figure 8 (panel a) suggests that protein clusters and vOTUs captured most of the diversity in the samples. However, there is no description of how protein clusters were determined, only a reference. And viral identifications were limited using strict criteria to identify virus genes that were similar to viral genes in other known viruses. Both methods are prone to systematic errors, in that identifications of new protein clusters and viruses are limited to what is in the published search database.

Response: We have provided the detailed method for the generation of viral protein clusters in Lines 806-813. (i) Our previous method used to generate viral protein clusters was based on the sequence similarity with known viral genomes (NR viral RefSeq database). Indeed, this method is limited to identifying viral protein clusters that are similar to known viral genomes in the NR viral RefSeq database. In order to identify bona fide viral protein clusters as much as possible, we re-performed the identification of viral protein clusters in seamount sediment viromes using GOV2.0 and IMG/VR database as described by Jian et al. (Lines 806-813). The new method identified much more viral protein clusters than the previous method (790,756 vs 19,761 PCs), and the results of the new analysis also support our conclusion that viruses in seamount sediments have a high degree of endemism (Lines 536-544). (ii) For the identification of viruses, we used a combination of three pipelines (Lines 701-720 and Supplementary Fig. 3a), and these pipelines have been widely used for the identification of viral sequences (Cheng R et al, 2022; Li Z et al, 2021; Luo X Q et al, 2022). In fact, these pipelines identify not only viral sequences similar to known viral genomes, but also identify novel viral sequences based on several virus-specific features (e.g., the presence of viral hallmark genes) (Nayfach S et al, 2021; Guo J et al, 2021; Kieft K et al, 2020).

Reference

Jian H, Yi Y, Wang J, et al. Diversity and distribution of viruses inhabiting the deepest ocean on Earth[J]. *The ISME journal*, 2021, 15(10): 3094-3110.

Cheng R, Li X, Jiang L, et al. Virus diversity and interactions with hosts in deep-sea hydrothermal vents[J]. *Microbiome*, 2022, 10(1): 1-17.

Li Z, Pan D, Wei G, et al. Deep sea sediments associated with cold seeps are a subsurface reservoir of viral diversity[J]. *The ISME Journal*, 2021, 15(8): 2366-2378.

Luo X Q, Wang P, Li J L, et al. Viral community-wide auxiliary metabolic genes differ by lifestyles, habitats, and hosts[J]. *Microbiome*, 2022, 10(1): 1-18.

Nayfach S, Camargo A P, Schulz F, et al. CheckV assesses the quality and completeness of metagenome-assembled viral genomes[J]. *Nature biotechnology*, 2021, 39(5): 578-585.

Guo J, Bolduc B, Zayed A A, et al. VirSorter2: a multi-classifier, expert-guided approach to detect diverse DNA and RNA viruses[J]. *Microbiome*, 2021, 9: 1-13.

Kieft K, Zhou Z, Anantharaman K. VIBRANT: automated recovery, annotation and curation of microbial viruses, and evaluation of viral community function from genomic sequences[J]. *Microbiome*, 2020, 8(1): 1-23.

Comment#6: Lines 344-347: The authors state that 97% of all virus hosts (1,953) were predicted from MAGs in GTDB and that all remaining hosts were from seamount sediment MAGs generated in their study, with no MAGs identified as hosts from other sediment samples. Does this mean that only 60 sediment MAGs were potential hosts? This seems like a relatively small number to make the broad statement that this implies endemism across different seamount habitats. Aren't seamount habitats under sampled?

Response: It is a good question. Yes, seamount habitats are relatively under-sampled, because we can obtain only 151 seamount MAGs from the IMG/M database, although they already represent the largest seamount MAG dataset currently available. These 151 seamount MAGs are assembled from several different seamount environments by several studies (Supplementary Table 12). Despite the limited number of seamount MAGs used for host prediction, there are notable differences in the

prediction frequency between MAGs generated in this study and MAGs from other seamount sediment samples. Fifty-nine of 63 MAGs assembled in this study were predicted hosts for viruses, whereas none of 151 MAGs from other seamount sediment samples were predicted hosts. In our opinion, this notable divergence should support our claim that “This result implies the potential endemism of viruses across different seamount habitats”.

Comment7#: The author’s show data which suggest that very different virus communities are identified depending on the method used to sequence them, such as a metagenome versus a metavirome. Can samples be compared across different seamounts? If so, how many studies have been conducted? Do they have similar depth of coverage?

Response: To the best of our knowledge, this is the first paper assessing viral communities in deep-sea seamount sediments using either metagenome or virome. Thus, there is no relevant study on seamount viruses that could be compared with our study. In fact, we only utilized MAGs assembled from other seamount samples for the prediction of host-virus linkages. Thus, the issue of similar depth of coverage is not relevant to this analysis.

Comment#8: Much is missing from the methods:

1. Sediment selection, collection, and characterization details are missing
2. DNA extraction methods are missing.
3. Details about viral purification. Many bacteria will pass through a 0.45 micron mesh. Was bacterial DNA identified in the metaviromes? If so, how much of it was bacterial and how was it removed?

Response: The detailed information for sediment sample selection, collection, and characterization has been provided in Lines 625-639. The information for DNA extraction methods has been provided in Lines 643-644. Besides, the detailed information for viral purification has been provided in Lines 681-690.

Yes, bacterial DNA was frequently identified in the viromes. However, bacterial DNA was removed for further analysis, since our pipelines used strict criteria to identify viral sequences from both virome and bulk metagenome (Lines 701-720 and

Supplementary Fig. 3a), and no host contamination should be included for the viral analysis.

Reviewer 2:

Comment#1: This study uses metagenomic and viromic sequencing and bioinformatic efforts to explore how microbial and viral community diversity, virus-host relationships, and viral ecological roles differ between various seamounts. The authors find extensive viral diversity and identify several viral-host relationships. AMGs encoded by viruses may ultimately play a role in influencing host metabolism and affect carbon, nitrogen, and sulfur cycling in these sediments while also enabling host survival in these environments. Authors also reveal that no hosts were predicted from MAGs assembles from other seamount sediments suggesting endemism of viral communities within seamounts. I thoroughly enjoyed reading this paper. I feel that this paper went above and beyond by analyzing prokaryotic and viral communities in seamount sediments, identifying viral diversity, virus-host linkages, and viral lifestyles, impacts of viruses on host metabolism, and searching for impacts of seamount features on the presence of local viral communities. In some cases, I think the analyses went beyond the current capabilities/knowledge base of the field and potentially made some bold statements that I feel can't be justified (i.e., classifying viral taxonomy based on a somewhat conserved viral gene). Overall, the work does support the conclusion and the claims. The methods are sound and thorough. I do believe this is the first paper assessing viral diversity and ecological roles in deep-sea seamount sediments, which is extremely exciting. I have made some specific comments on areas where I would improve the paper, but ultimately these notes are more focused on removing or rephrasing information as opposed to adding additional analyses.

Response: Thanks very much for your positive feedbacks. Your comments have all been most valuable and helpful for improving the manuscript. The manuscript has been extensively revised according to your comments as listed below. All the changes in this revision are marked in Red color for easy recognition.

Comment#2: The introduction holds a lot of great information, but the first two

paragraphs could be made more concise.

Response: We have revised the first two paragraphs to make them more concise as suggested (Lines 75-93).

Comment#3: It would be good to include citations for the statement made in Line 133.

Response: We have added references for this statement in Line 124

Comment#4: Line 190-194: This seems a bit obvious. I would remove “as several studies have suggested.” I think it should be reworded because these are two very different methods and viral recovery is not coincidentally better if you are enriching for viruses. Typically, metagenomes are acquired by filtering onto 0.2um filters and therefore viruses slip through and are not accounted for during sequencing (with the exception of temperate phage and viruses that are actively infecting the cellular fraction). I think this is made clear towards the end of the sentence, so I would remove “as several studies have suggested” because this really is common practice and knowledge.

Response: We have revised the sentence as suggested in Lines 184-187.

Comment#5: T Line 208: “to huge viruses” I believe they are more commonly referred to as “giant viruses” ?

Response: We have revised the term “huge viruses” to “giant viruses” as suggested in Line 201.

Comment#6: Line 239: Couldn't this also be because these viruses may be temperate or actively infecting the hosts (i.e., are not ambient and wouldn't otherwise be detected in the free viral population)?

Response: This possibility has also been discussed in the text in L238-240.

Comment#7: Figure 3e: This is a great figure.

Response: Thanks very much for your positive feedback.

Comment#8: Line 254: Might be worth noting the bias that the viral database itself plays in recognizing Caudovirales viruses (since these makes up a majority of the viral database). It therefore makes sense that the viral database continues to largely identify Caudovirales.

Response: This note has been added as suggested in Lines 254-257.

Comment#9: Lines 261-267: I don't think that placing viral sequences in a phylogenetic tree and making inferences on lineages is important in the goal of this paper, and I don't think it is correct practice. (If it is, several papers need to be cited to support this method and claim.) It is complicated to assess viral phylogeny because they do not have conserved genes, and even if a gene is conserved, I'm not sure you can make claims regarding viral diversity based on this one gene. You also come to this conclusion/ show this in Lines 290-295.

Response: This is a very good question. We agree with you that making phylogenetic inferences based on one conserved viral gene may be problematic, since significant genetic divergence exists amongst viral subgroups. In addition, we also agree with you that placing viral sequences in a phylogenetic tree and making inferences on lineages is not our primary goal for this paper. Therefore, we have placed Fig. 4 into supplements (Supplementary Fig. 5), and deleted relevant claims on viruses forming different groups based on the phylogeny of a single gene (Lines 257-265 and 276-289).

Comment#10: Figure 4: It is very hard to tell which are the reference genes as the grey circles are too small in 4a. I would find a way to make this more obvious. I also feel that more justification is needed in order to try to classify different viruses based on a shared gene. Figure 4a doesn't seem to mean very much as it is now. Are you trying to define taxonomy based on a gene? If so this should be its own paper in itself. I would either remove Figure 4 or move it to the supplemental. I would also be more clear on what your goal is in trying to map viral phylogeny. If it is for viral diversity, then I would be careful not to make claims about viruses forming different groups based on a gene that really can't be used as an identifier since they are not conserved. Instead you can focus on Figure 4b and talk about how genome arrangements are different, but again this is expected, so consider moving this to the supplements.

Response: This is a very good question. We agree with you that making phylogenetic inferences based on one conserved viral gene may be problematic, since significant genetic divergence exists amongst viral subgroups. In addition, we also

agree with you that placing viral sequences in a phylogenetic tree and making inferences on lineages is not our primary goal for this paper. Therefore, we have placed Fig. 4 into supplements (Supplementary Fig. 5), and deleted relevant claims on viruses forming different groups based on the phylogeny of a single gene (Lines 257-265 and 276-289). Moreover, we have revised Fig. 4a (now Supplementary Fig. 5) to mark reference genes with black dots, which will make them more obvious.

Comment#11: Figure 5a: The text could explain what we are looking at a bit better. The caption may want to include the organization of nodes from greater to fewer.

Response: For easy understanding of Fig. 5a (now Fig. 4a), the positions of seamount viruses are marked with black circles. In addition, the caption has been organized according to the node number from greater to fewer as suggested.

Comment#12: Figure 5b: The two numbered scales to the left need labels to be more clear. I love this figure.

Response: We have added labels to the two scales on the left in Fig. 5b (now Fig. 4b) as suggested.

Comment#13: Line 330 & 343: The full name is GTDB-tk and should be used as such.

Response: We have revised the phrase as suggested throughout the manuscript.

Comment#14: Line 346: Very interesting!

Response: Thanks very much for your positive feedback.

Comment#15: Line 607: I believe VLPs refer to viral like particles, not necessarily ambient or free viruses. I would describe these as ambient viruses instead of VLPs.

Response: We have replaced the phrase VLPs with ambient viruses throughout the manuscript as suggested.

Comment#16: Overall, I would absolutely recommend this paper for publishing.

Response: Thanks very much for your positive feedback.

REVIEWER COMMENTS

Reviewer #1 (Remarks to the Author):

The authors have not adequately addressed my key concerns regarding their conclusion that endemism among viruses in sediments is a key finding from this research. They have added a supplemental table that provides some additional information about the sediment samples that were collected and they have analyzed OTUs in a new way that is presented in a supplemental figure that compares shifts in community structure of both prokaryotes and viruses with measured environmental variables such as carbon, nitrogen, and depth. I appreciate the new information but feel that the evidence for endemism is insufficient. This is primarily because 1, there are not enough well characterized sediment types and not enough geological or oceanographic data to support this conclusion and 2, the data in the new supplemental figure (2) and main figure (7) do not agree.

1. Differences in sediment sample types are not statistically robust. I would expect to see replicates from different types of well characterized sediments and or geological or oceanographic features to draw the endemism conclusion.

2. The RDA analyses suggest that communities are similar in very distant samples, and that they are similar for both prokaryotes and viruses. For example, NP4_S05 and NA_S06 group together relative to all the other samples, for prokaryotes along axis RDA2 and for viruses along axis RDA1. Similarly, the vectors for carbon, nitrogen, and depth indicate positive correlations in the direction of NP4_S05 and NA_S06 for both prokaryotes and viruses. Similarly, NLG_S05 groups separately for prokaryotes along axis RDA1 and for viruses along axis RDA2. Axes in these types of plots can be rotated. If the prokaryote plot is rotated counterclockwise so that the variables on axes switches, it is very similar to the plot for viruses, along both axes and with the measured environmental variables. To me, this suggests that prokaryote and virus communities are similar in samples driven by nutrient concentrations and rather than geography.

More data characterizing more sediment types, geology, or physical oceanography is really needed to draw these types of conclusions, especially the idea of endemism. However, a robust characterization of prokaryotes and viruses in diverse seamount sediment types is warranted.

The methods are still lacking in detail. Very little information is provided for the RDA analyses, for example. Were the data normalized, how many species (OTUs) were used in each analysis, were outlier analyses conducted, what distance measure was used, what was the final stress or P value . . .

Reviewer #2 (Remarks to the Author):

Authors have addressed all questions and comments that were previously made. The addition of the RDA analysis and details on DNA extraction and viral purification methods improved the manuscript. I do worry about DNA extraction and amplification bias towards prokaryotes if you are simply removing metagenomes from the sequenced "virome" that passed through 0.45um. This is very much not common practice (the virome would typically be acquired from 0.2um filtrate or collected onto 0.02um filters) and it should be addressed how this method could impact virome recovery and results.

Minor comments on text:

Line 90: Year for Myrna et al. not provided.

Line 93: Year for Huo et al. not provided.

Line 257: Typo: "the" should be "then"

Responses to reviewers' comments (point-to-point)

Reviewer 1:

Comment#1: The authors have not adequately addressed my key concerns regarding their conclusion that endemism among viruses in sediments is a key finding from this research. They have added a supplemental table that provides some additional information about the sediment samples that were collected and they have analyzed OTUs in a new way that is presented in a supplemental figure that compares shifts in community structure of both prokaryotes and viruses with measured environmental variables such as carbon, nitrogen, and depth. I appreciate the new information but feel that the evidence for endemism is insufficient. This is primarily because 1, there are not enough well-characterized sediment types and not enough geological or oceanographic data to support this conclusion and 2, the data in the new supplemental figure (2) and main figure (7) do not agree.

Response: Thanks very much for your comments, which have all been most valuable and helpful for improving the manuscript. The manuscript has been extensively revised according to your comments, which are listed below. All the changes in this revision are marked in Red color for easy recognition.

It is a very good question. After careful consideration and evaluation, we agree that our study could not adequately support our previous bold conclusion on viral endemism among seamount sediments. Indeed, more data characterizing more sediment types, geology, or physical oceanography is needed to draw the bold conclusion of viral endemism. Therefore, we have removed the claims of viral endemism throughout the manuscript; instead, we have rephrased relevant sentences to tone down the claims, *i.e.*, high divergence of viral communities across seamounts. The revisions are in the Title, Abstract section (Lines 40-42), Results and Discussion section (Lines 343-344, 534-536, 543-552, 566-568, 588-591), and Conclusion section (Lines 606-609). In addition, we have provided more discussions to point out the need for more well-characterized sediments and geological/oceanographic data in

the future to verify our hypothesis (Lines 591-595). As for concerns raised about the RDA analysis, please refer to the response to Comment#3 below.

Comment#2: 1. Differences in sediment sample types are not statistically robust. I would expect to see replicates from different types of well characterized sediments and or geological or oceanographic features to draw the endemism conclusion.

Response: Indeed, more data characterizing more sediment types, geology, or physical oceanography is needed to draw the bold conclusion of viral endemism. Therefore, we have removed the claims of viral endemism throughout the manuscript; instead, we have rephrased relevant sentences to tone down the claims, *i.e.*, high divergence of viral communities across seamounts. The revisions are in the Title, Abstract section (Lines 40-42), Results and Discussion section (Lines 343-344, 534-536, 543-552, 566-568, 588-591), and Conclusion section (Lines 606-609). In addition, we have provided more discussions to point out the need for more well-characterized sediments and geological/oceanographic data in the future to verify our hypothesis (Lines 591-595). As for concerns raised about the RDA analysis, please refer to the response to Comment#3 below.

Comment#3: The RDA analyses suggest that communities are similar in very distant samples, and that they are similar for both prokaryotes and viruses. For example, NP4_S05 and NA_S06 group together relative to all the other samples, for prokaryotes along axis RDA2 and for viruses along axis RDA1. Similarly, the vectors for carbon, nitrogen, and depth indicate positive correlations in the direction of NP4_S05 and NA_S06 for both prokaryotes and viruses. Similarly, NLG_S05 groups separately for prokaryotes along axis RDA1 and for viruses along axis RDA2. Axes in these types of plots can be rotated. If the prokaryote plot is rotated counterclockwise so that the variables on axes switches, it is very similar to the plot for viruses, along both axes and with the measured environmental variables. To me, this suggests that prokaryote and virus communities are similar in samples driven by nutrient concentrations and rather than geography.

Response: Thanks very much for pointing out the problems with the Redundancy Analysis (RDA) analysis. To evaluate the robustness of the RDA model, we

performed a permutation test. The detailed method for RDA analysis and permutation test is provided at the end of this Response. As shown in Table R1 below, the *p* values for the RDA axes and the entire RDA model for viruses are very large (>0.5), and the *p* values for the RDA axes 2 and 3 for prokaryotes are >0.3. This result indicates that these RDA models are not reliable and, thus, should be removed. Sorry for our mistakes. We have replaced the RDA analysis with Mantel's correlation analysis for a robust evaluation of the correlation of environmental variables with prokaryote and virus communities. The detailed method for Mantel's correlation analysis has been provided in the Methods and Materials section in Lines 729-737. The relevant data and descriptions for Mantel's correlation analysis are provided in Supplementary Fig. 2 and Lines 155-159 and 215-217. The detailed values for the Mantel's correlation analysis are listed in Table R2 below.

Table R1 Permutation test to evaluate the robustness of the RDA model

	Prokaryotes		Viruses	
	p	Sig.	p	Sig.
RDA axis 1	0.03333	Y	0.7986	N
RDA axis 2	0.38333	N	0.9417	N
RDA axis 3	0.5125	N	0.6819	N
Model	0.03056	Y	0.9	N

Table R2 Correlation analysis between environmental variables and prokaryotic and viral OTU profiles by the Mantel test

OTU	Environmental variable	r	p	rd
Prokaryotes	Depth	0.678164	0.041667	>0.4
Prokaryotes	Organic carbon	0.201183	0.159722	0.2 - 0.4
Prokaryotes	Nitrogen	0.730804	0.0625	> 0.4
Viruses	Depth	-0.59131	0.9375	< 0.2
Viruses	Organic carbon	-0.04508	0.522222	< 0.2
Viruses	Nitrogen	-0.63308	0.902778	< 0.2

Detailed method for RDA analysis and permutation test

Prokaryotic and viral OTU abundance data were Hellinger transformed using the R package *vegan* v2.6-4 (function `decostand`) (Dixon P, 2003) in order to reduce the weight of abundant OTUs while preserving Euclidean distances between samples in the multidimensional space (Legendre P & Gallagher E D, 2001). A preliminary detrended correspondence analysis performed on the transformed OTU data revealed a gradient length of 3.22 (for prokaryotes) and 1.74 (for viruses). We performed RDA analysis using the function `rda` and tested the significance using the function `anova`. Selection was performed using the function `ordiR2step` with permutation test (999 permutations) in the R package *vegan* v2.6-4 (Dixon P, 2003). The selected variables were then used as explanatory variables for RDA, and ANOVA was run with 999 permutations to assess the significance of constraints using the function `anova` in the R package *vegan* v2.6-4 (Dixon P, 2003).

Reference:

Dixon P. VEGAN, a package of R functions for community ecology[J]. *Journal of Vegetation Science*, 2003, 14(6): 927-930.

Legendre P, Gallagher E D. Ecologically meaningful transformations for ordination of species data[J]. *Oecologia*, 2001, 129: 271-280.

Comment#4: More data characterizing more sediment types, geology, or physical oceanography is really needed to draw these types of conclusions, especially the idea of endemism. However, a robust characterization of prokaryotes and viruses in diverse seamount sediment types is warranted.

Response: Indeed, more data characterizing more sediment types, geology, or physical oceanography is needed to draw the bold conclusion of viral endemism. Therefore, we have removed the claims of viral endemism throughout the manuscript; instead, we have rephrased relevant sentences to tone down the claims, *i.e.*, high divergence of viral communities across seamounts. The revisions are in the Title, Abstract section (Lines 40-42), Results and Discussion section (Lines 343-344, 534-536, 543-552, 566-568, 588-591), and Conclusion section (Lines 606-609). In addition, we have provided more discussions to point out the need for more well-

characterized sediments and geological/oceanographic data in the future to verify our hypothesis (Lines 591-595). As for concerns raised about the RDA analysis, please refer to the response to Comment#3 below.

Comment#5: The methods are still lacking in detail. Very little information is provided for the RDA analyses, for example. Were the data normalized, how many species (OTUs) were used in each analysis, were outlier analyses conducted, what distance measure was used, what was the final stress or P value . . .

Response: Since the RDA analysis is replaced with the Mantel's correlation analysis, the RDA method has been removed from the manuscript. Nevertheless, the detailed RDA method is provided below for your reference. Besides, the detailed method for Mantel's correlation analysis has been provided in the Methods and Materials section in Lines 729-737.

Detailed method for RDA analysis and permutation test

Prokaryotic and viral OTU abundance data were Hellinger transformed using the R package *vegan* v2.6-4 (function `decostand`) (Dixon P, 2003) in order to reduce the weight of abundant OTUs while preserving Euclidean distances between samples in the multidimensional space (Legendre P & Gallagher E D, 2001). A preliminary detrended correspondence analysis performed on the transformed OTU data revealed a gradient length of 3.22 (for prokaryotes) and 1.74 (for viruses). We performed RDA analysis using the function `rda` and tested the significance using the function `anova`. Selection was performed using the function `ordiR2step` with permutation test (999 permutations) in the R package *vegan* v2.6-4 (Dixon P, 2003). The selected variables were then used as explanatory variables for RDA, and ANOVA was run with 999 permutations to assess the significance of constraints using the function `anova` in the R package *vegan* v2.6-4 (Dixon P, 2003).

Reference:

Dixon P. VEGAN, a package of R functions for community ecology[J]. *Journal of Vegetation Science*, 2003, 14(6): 927-930.

Legendre P, Gallagher E D. Ecologically meaningful transformations for ordination of species data[J]. *Oecologia*, 2001, 129: 271-280.

Reviewer 2:

Comment#1: Authors have addressed all questions and comments that were previously made. The addition of the RDA analysis and details on DNA extraction and viral purification methods improved the manuscript. I do worry about DNA extraction and amplification bias towards prokaryotes if you are simply removing metagenomes from the sequenced “virome” that passed through 0.45µm. This is very much not common practice (the virome would typically be acquired from 0.2µm filtrate or collected onto 0.02µm filters) and it should be addressed how this method could impact virome recovery and results.

Response: Thanks very much for your comments, which have all been most valuable and helpful for improving the manuscript. The manuscript has been extensively revised according to your comments, which are listed below. All the changes in this revision are marked in Red color for easy recognition.

It is a very good question. Indeed, viromes were initially typically acquired from 0.2 µm filtrate (Breitbart M et al., 2002; 2004). However, the discoveries of viruses with increasingly larger capsid sizes necessitate the re-evaluation of common filtration methods when attempting to capture entire virus communities (Claverie J M et al., 2018). It is found that the 0.2-µm-size fraction excludes many diverse large viruses (Göller P C et al., 2020; Conceição-Neto N et al., 2015). For example, in one study of soil virome, the use of 0.45-µm-pore-size filters approximately doubled the number of observable virus-like particles compared to that with the 0.2-µm-pore-size filters (Göller P C et al., 2020). In particular, many nucleocytoplasmic large DNA viruses (NCLDV) can be excluded from the virome obtained by 0.22-µm filtration, and the virome obtained by 0.45-µm filtration has the potential for higher giant viruses recovery (Göller P C et al.,2020; Conceição-Neto N et al., 2015). In this context, more and more studies have opted for the use of 0.45-µm-pore-size filters in virome research in recent years. (Vibin J et al, 2018; Deng L et al, 2019; Bekliz M et al, 2019; Lachnit et al, 2019; Adriaenssens et al, 2021). In addition, several studies indicate the pervasive presence of non-viral sequence contamination in viromes, no matter whether a 0.2 µm or 0.45 µm filter is used (Perlejewski K et al, 2015; Moustafa

A et al, 2017; Asplund M et al, 2019; Jurasz H et al, 2021). The non-viral sequence contamination in viromes seems challenging to avoid (Zolfo M et al., 2019). Nevertheless, our powerful pipelines used strict criteria to identify viral sequences from virome (Lines 701-720 and Supplementary Fig. 3a), and no host contamination should be included in the viral analysis. To make the manuscript clearer, we have added some brief discussions on the impact of 0.45- μ m filtration on virome recovery and results (Lines 687-690).

Reference

- Adriaenssens E M, McDonald J E, Jones D L, et al. Tracing the fate of wastewater viruses reveals catchment-scale virome diversity and connectivity[J]. *Water Research*, 2021, 203: 117568.
- Asplund M, Kjartansdóttir K R, Mollerup S, et al. Contaminating viral sequences in high-throughput sequencing viromics: a linkage study of 700 sequencing libraries[J]. *Clinical Microbiology and Infection*, 2019, 25(10): 1277-1285.
- Bekliz M, Brandani J, Bourquin M, et al. Benchmarking protocols for the metagenomic analysis of stream biofilm viromes[J]. *PeerJ*, 2019, 7: e8187.
- Breitbart M, Felts B, Kelley S, et al. Diversity and population structure of a near-shore marine-sediment viral community[J]. *Proceedings of the Royal Society of London. Series B: Biological Sciences*, 2004, 271(1539): 565-574.
- Breitbart M, Salamon P, Andresen B, et al. Genomic analysis of uncultured marine viral communities[J]. *Proceedings of the National Academy of Sciences USA*, 2002, 99(22): 14250-14255.
- Claverie J M, Abergel C. Mimiviridae: an expanding family of highly diverse large dsDNA viruses infecting a wide phylogenetic range of aquatic eukaryotes[J]. *Viruses*, 2018, 10(9): 506.
- Conceição-Neto N, Zeller M, Lefrère H, et al. Modular approach to customise sample preparation procedures for viral metagenomics: a reproducible protocol for virome analysis[J]. *Scientific reports*, 2015, 5(1): 16532.
- Deng L, Silins R, Castro-Mejía J L, et al. A protocol for extraction of infective viromes suitable for metagenomics sequencing from low volume fecal samples[J].

Viruses, 2019, 11(7): 667.

Göller P C, Haro-Moreno J M, Rodriguez-Valera F, et al. Uncovering a hidden diversity: optimized protocols for the extraction of dsDNA bacteriophages from soil[J]. Microbiome, 2020, 8: 1-16.

Jurasz H, Pawłowski T, Perlejewski K. Contamination issue in viral metagenomics: problems, solutions, and clinical perspectives[J]. Frontiers in Microbiology, 2021, 12: 745076.

Lachnit T, Dafforn K A, Johnston E L, et al. Contrasting distributions of bacteriophages and eukaryotic viruses from contaminated coastal sediments[J]. Environmental Microbiology, 2019, 21(6): 1929-1941.

Moustafa A, Xie C, Kirkness E, et al. The blood DNA virome in 8,000 humans[J]. PLoS pathogens, 2017, 13(3): e1006292.

Perlejewski K, Popiel M, Laskus T, et al. Next-generation sequencing (NGS) in the identification of encephalitis-causing viruses: unexpected detection of human herpesvirus 1 while searching for RNA pathogens[J]. Journal of Virological Methods, 2015, 226: 1-6.

Vibin J, Chamings A, Collier F, et al. Metagenomics detection and characterisation of viruses in faecal samples from Australian wild birds[J]. Scientific Reports, 2018, 8(1): 8686.

Zolfo M, Pinto F, Asnicar F, et al. Detecting contamination in viromes using ViromeQC[J]. Nature Biotechnology, 2019, 37(12): 1408-1412.

Comment#2: Line 90: Year for Myrna et al. not provided.

Response: We have provided the year for Myrna et al. in Line 90.

Comment#3: Line 93: Year for Huo et al. not provided.

Response: We have provided the year for Huo et al. in Line 93.

Comment#4: Line 257: Typo: “the” should be “then”

Response: We have corrected the sentence in Line 259.

REVIEWERS' COMMENTS

Reviewer #1 (Remarks to the Author):

I am satisfied with the major changes the authors made to this manuscript.

Reviewer #2 (Remarks to the Author):

All previous comments have been adequately addressed.